# The autism- and schizophrenia-associated protein CYFIP1 regulates bilateral brain connectivity and behaviour

Nuria Domínguez-Iturza [1,2], Adrian C. Lo[1], Disha Shah[3,4], Marcelo Armendáriz [5], Anna Vannelli[1], Valentina Mercaldo[1], Massimo Trusel[1], Ka Wan Li [6], Denise Gastaldo[1], Ana Rita Santos[2,7], Zsuzsanna Callaerts-Vegh [8], Rudi D'Hooge[8], Manuel Mameli [1], Annemie Van der Linden[3], August B. Smit[6], Tilmann Achsel[1,2] & Claudia Bagni [1,2,9]

Copy-number variants of the *CYFIP1* gene in humans have been linked to autism spectrum disorders (ASD) and schizophrenia (SCZ), two neuropsychiatric disorders characterized by defects in brain connectivity. Here, we show that CYFIP1 plays an important role in brain functional connectivity and callosal functions. We find that *Cyfip1*-heterozygous mice have reduced functional connectivity and defects in white matter architecture, similar to phenotypes found in patients with ASD, SCZ and other neuropsychiatric disorders. *Cyfip1*-deficient mice also present decreased myelination in the callosal axons, altered presynaptic function, and impaired bilateral connectivity. Finally, *Cyfip1* deficiency leads to abnormalities in motor coordination, sensorimotor gating and sensory perception, which are also known neuropsychiatric disorder-related symptoms. These results show that *Cyfip1* haploinsufficiency compromises brain connectivity and function, which might explain its genetic association to neuropsychiatric disorders.

[1] Department of Fundamental Neurosciences, University of Lausanne, 1005 Lausanne, Switzerland. [2] Department of Human Genetics KU Leuven, VIB Center for Brain & Disease Research, 3000 Leuven, Belgium. [3] Department of Biomedical Sciences, Bio-Imaging Laboratory, University of Antwerp, 2610 Antwerp, Belgium. [4] Department of Neuroscience KU Leuven, VIB Center for Brain & Disease Research, 3000 Leuven, Belgium. [5] Department of Neurosciences, Laboratory of Neuro- and Psychophysiology, KU Leuven, 3000 Leuven, Belgium. [6] Department of Molecular and Cellular Neurobiology, Center for Neurogenomics and Cognitive Research, VU University Amsterdam, 1081 Amsterdam, The Netherlands. [7] VIB Discovery Sciences, Bioincubator, 3001 Heverlee, Belgium. [8] Faculty of Psychology and Educational Sciences, KU Leuven, Laboratory of Biological Psychology, 3000 Leuven, Belgium. [9] Department of Biomedicine and Prevention, University of Rome Tor Vergata, 00133 Rome, Italy. Correspondence and requests for materials should be addressed to C.B. (email: Claudia.Bagni@unil.ch)

The corpus callosum (CC) is the largest axonal commissure in the human brain and connects both cerebral hemispheres. Appropriate callosal microstructure and function are required for bilateral functional connectivity[1,2]. Agenesis of the corpus callosum (ACC), callosotomy or defects in callosal integrity have been linked to deficits in social communication[3], social behaviour[4], motor coordination and learning[5,6], and cognitive performance[7]. Similarly, CC abnormalities have been found in patients with diverse neuropsychiatric disorders such as autism spectrum disorders (ASD), schizophrenia (SCZ), fragile X syndrome (FXS), attention-deficit hyperactivity disorder (ADHD), Prader–Willi syndrome and 22q11.2 deletion syndrome[3,8–11]. Interestingly, over the last decade, magnetic resonance imaging (MRI) studies in patients with ASD and SCZ have identified functional connectivity and microstructural defects in the white matter as a hallmark of these disorders. Previous work based on resting-state functional MRI (rsfMRI) revealed that brain functional connectivity (FC) is altered in ASD and SCZ patients, showing predominantly reduced long-range functional connectivity[12–15]. Additionally, recent studies using diffusion tensor imaging (DTI) showed that axonal integrity and connectivity are also affected in patients with ASD and SCZ[16–18]. In particular, DTI revealed abnormalities in the corpus callosum of these patients[16,19]. Furthermore, reduced myelination has also been observed in patients with ASD and SCZ[20,21].

Recently, copy-number variations (CNVs) of the chromosomal region 15q11.2 have been associated with the development of neuropsychiatric disorders, especially ASD and SCZ, and domain-specific cognitive impairments[22]. The proximal region of the long arm of chromosome 15 contains several breakpoints, resulting in different rearrangements[23]. The smallest region that has been linked to neuropsychiatric disorders is contained between the breakpoints BP1 and BP2, i.e., the 15q11.2 locus. CNVs of this region are quite frequent (~1% of the population[24]) and cause a small but significant increase in the risk of developing ASD or SCZ[25,26]. In all, 15q11.2 deletion and duplication carriers show abnormal volume of the corpus callosum among other cerebral and white matter abnormalities[27–30]. The chromosomal region contained in the BP1–BP2 encodes four genes: *TUBGCP5*, *NIPA1*, *NIPA2* and *CYFIP1*. Rare CNVs, single-nucleotide polymorphisms (SNPs) and point mutations single out *CYFIP1* as the gene of that region most likely associated with SCZ[31–33] and ASD[34,35]. CYFIP1, the Cytoplasmic FMRP interacting protein 1, first described as "specifically Rac1-associated" (Sra-1) protein[36], is part of the Wave Regulatory Complex (WRC), a heteropentamer formed by WAVE1/2/3, CYFIP1, ABI1/2, NCKAP1 and HPSC300[37,38]. The WRC functions as an Arp2/3 regulator, promoting its actin-nucleating activity, and therefore controlling actin assembly[37]. This process can be activated by Rac1 signalling, which promotes a conformational change in CYFIP1[39] and its binding to the WRC[38]. In its alternative conformation, CYFIP1 binds FMRP and the cap-binding protein eIF4E, repressing protein translation of specific mRNAs[40]. Both processes, actin dynamics and regulated protein synthesis, are crucial for synaptic development and brain functioning, strengthening the hypothesis that alterations in the *CYFIP1* gene can account for the clinical features of patients with ASD and SCZ.

Deletion of the *Cyfip1* gene in mice leads to embryonic lethality[38,41,42] while heterozygous mice show aberrant behaviour[41,43,44], decreased dendritic complexity and increased number of immature spines[38,42,45]. *Cyfip1* heterozygous mice also present electrophysiological defects, such as enhanced mGluR-dependent LTD[41] and vesicle release probability[46]. These observations suggest that *Cyfip1* heterozygous mice are a suitable model for the neurological defects observed in BP1–BP2 haploinsufficient patients. Recently, it was shown in zebrafish that

CYFIP1 regulates axonal growth of retinal ganglion cells (RGSs)[47] and *Cyfip1* overexpression in mice leads to an increased proportion of mature spines[48]. The role of *Cyfip1* in brain connectivity, however, remains unexplored. Here, we show that *Cyfip1* heterozygous mice (*Cyfip1*[+/−]) have reduced functional connectivity and white-matter architecture defects. *Cyfip1* haploinsufficiency leads to a decreased transmission of callosal synapses, reduced myelination in the corpus callosum, and behaviourally manifests with deficits in motor coordination as well as other traits reminiscent of ASD and SCZ-like symptoms. Together, our findings suggest that *CYFIP1* is involved in the abnormal brain structures and callosal defects detected by MRI in individuals with the 15q11.2 deletion, and possibly in other forms of neuropsychiatric disorders with reduced CYFIP1 levels.

## Results

**Bilateral functional connectivity is impaired in *Cyfip1*[+/−] mice.** Many neuropsychiatric disorders are characterized by impaired brain connectivity. To investigate whether and how *Cyfip1* haploinsufficiency could affect brain networks in vivo, we analysed functional connectivity (FC) in several brain regions using resting-state functional magnetic resonance imaging (rsfMRI). *Cyfip1*[+/−] adult mice showed decreased FC compared to wild-type (WT) (compare the lower left and upper right halves of the matrix, Fig. 1a). Brain areas that presented significant difference between WT and *Cyfip1*[+/−] mice are the cingulate cortex (Cg) and the thalamus (T) (Fig. 1b and Supplementary Fig. 1a). FC networks are characterised by midline symmetry, and correlations between homotopic regions are particularly strong[49]. To analyse bilateral FC, seed-based analysis was performed across brain areas (Fig. 1c). Remarkably, FC with the contralateral side of the seed region was particularly affected in the hippocampus, somatosensory and motor cortices (Fig. 1d). In conclusion, our results show significant defects in functional connectivity, especially of the bilateral connections.

**Aberrant white matter architecture in *Cyfip1*[+/−] mice.** Bilateral connectivity mostly passes through the corpus callosum and abnormalities in this brain structure are regarded a hallmark of neuropsychiatric disorders. To determine whether the observed functional deficits are due to structural axonal abnormalities in the CC, we performed diffusion tensor imaging (DTI)[50] in WT and *Cyfip1*[+/−] animals. DTI is one of the most widely used techniques to analyse white matter architecture and integrity. Among the parameters obtained with DTI, fractional anisotropy (FA) is the most widely used. High FA values are indicative of fibre-like structures and therefore, FA provides an estimate of axonal architecture[50]. In Fig. 2a, the FA values across WT and *Cyfip1*[+/−] brains were colour-coded, with higher values in red highlighting white matter tracts such as the CC. The differential map of WT and *Cyfip1*[+/−] FA values showed that the highest reduction in FA was detected in the CC, in particular in the genu, its anterior part (Fig. 2b; first image). In addition, the statistical map showed that several brain regions have a nominally significant difference in FA between WT and *Cyfip1*[+/−] brains (Fig. 2c). Again, these results pointed to the corpus callosum as the most affected white matter tract. We therefore calculated the DTI parameters in the CC (Supplementary Fig. 1b) and no statistical differences were observed in the mean (MD), axial (AD), or radial diffusivity (RD). In contrast, fractional anisotropy was significantly lower in the CC of *Cyfip1*[+/−] mice (Fig. 2d), which might indicate changes in axonal thickness or myelination. Of note, we found a trend of increase (*t*-test with Holm-Sidak's correction, $p = 0.195$) in radial diffusivity, suggesting defects in myelination[50]. In support of this, we observed that the volume of

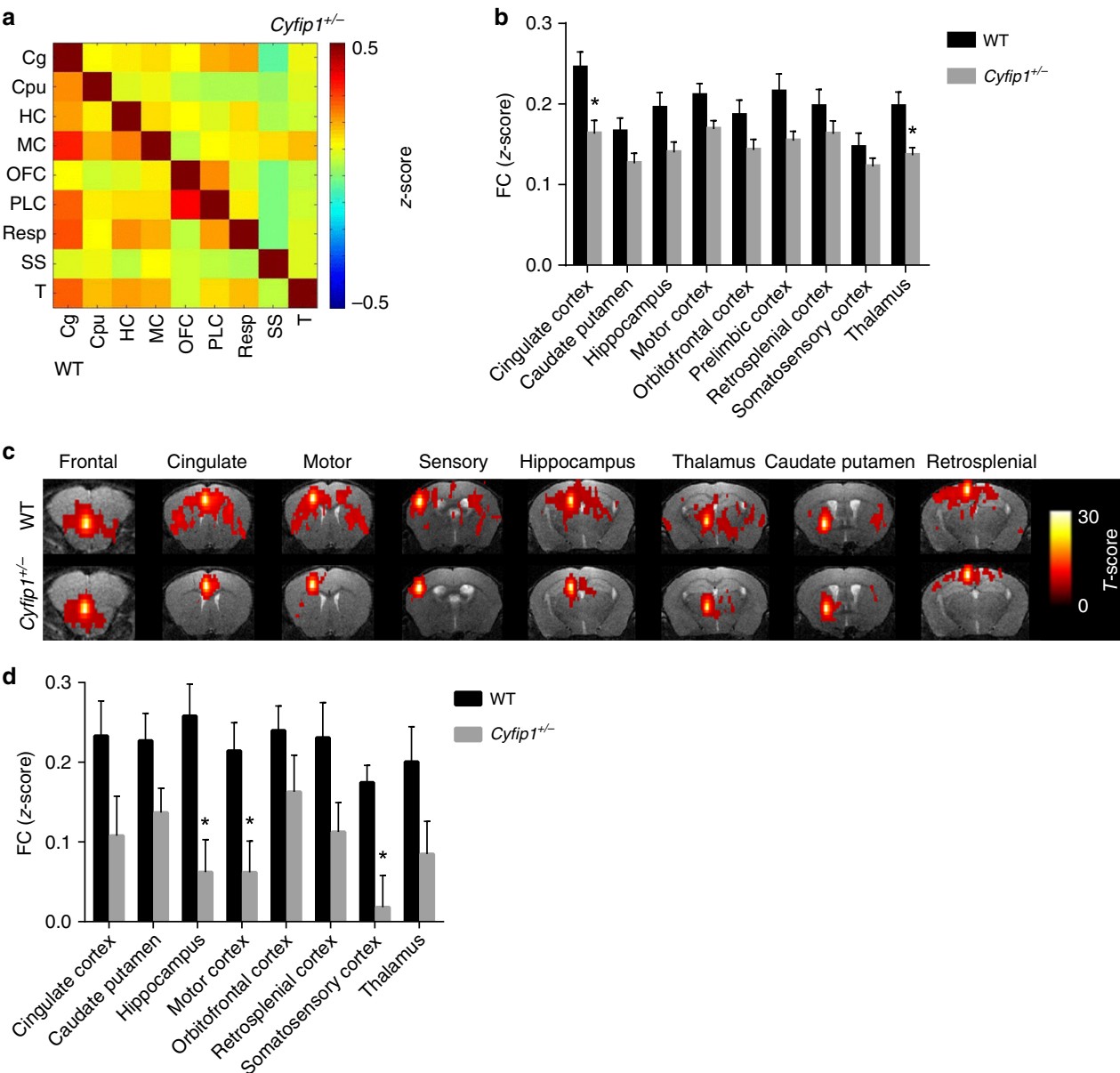

**Fig. 1** $Cyfip1^{+/-}$ mice show reduced functional connectivity. **a** Functional connectivity matrices of WT and $Cyfip1^{+/-}$ mice (lower and upper half of the matrix, respectively) (postnatal day 60, P60), in which functional correlation (z-score) between pairs of regions is represented by a colour scale (abbreviations: Cg = cingulate cortex, Cpu = caudate putamen, HC = hippocampus, MC = motor cortex, OFC = orbitofrontal cortex, PLC = prelimbic cortex, Resp = retrosplenial cortex, SS = somatosensory cortex, T = thalamus). **b** Based on the matrix in **a**, average functional connectivity strength of each region with all other regions is plotted for WT and $Cyfip1^{+/-}$ mice (WT $n = 15$ and $Cyfip1^{+/-}$ $n = 17$ mice; mean ± SEM; Holm-Sidak t-test; Cg, *$p = 0.0178$ and T, *$p = 0.0196$). **c** Seed-based analysis represented by functional connectivity maps for WT and $Cyfip1^{+/-}$ animals. The strength of connectivity for the seed region, indicated above each image, is mapped by a colour scale representing the T-score. **d** Average bilateral functional connectivity strength for selected brain regions in WT and $Cyfip1^{+/-}$ mice (WT $n = 15$ and $Cyfip1^{+/-}$ $n = 17$ mice; mean ± SEM; Holm-Sidak t-test; MC *$p = 0.0452$, SS *$p = 0.0157$, HC *$p = 0.0148$)

the corpus callosum was reduced in $Cyfip1^{+/-}$ mice compared to WT (Supplementary Fig. 1c).

To determine which axonal property (between axonal thickness and myelination) was affected, we performed electron microscopy of the corpus callosum. Representative micrographs from the anterior part of the CC were selected (Fig. 3a), and myelinated axons were automatically identified and parameterised. For each axon, the diameter and the myelin thickness were determined. In addition, the g-ratio, i.e., the ratio of the internal over the external axon diameter (Fig. 3b) was calculated. We observed no changes in the axonal diameter between the two genotypes, whereas the myelin thickness was reduced (Fig. 3c).

Consequently, the g-ratio increased across all axon sizes (Fig. 3c). In conclusion, $Cyfip1$ haploinsufficiency causes defects in functional connectivity, especially of the callosal projections that connect the two cortical hemispheres, which correlates with a reduced myelin thickness of the callosal fibres.

**$Cyfip1^{+/-}$ mice have altered callosal presynaptic function.** Changes in myelin thickness might be due to abnormal number of oligodendrocytes, which are the cells responsible for axonal myelination[51]. However, the number of mature (myelinating) and immature oligodendrocytes in the corpus callosum revealed no

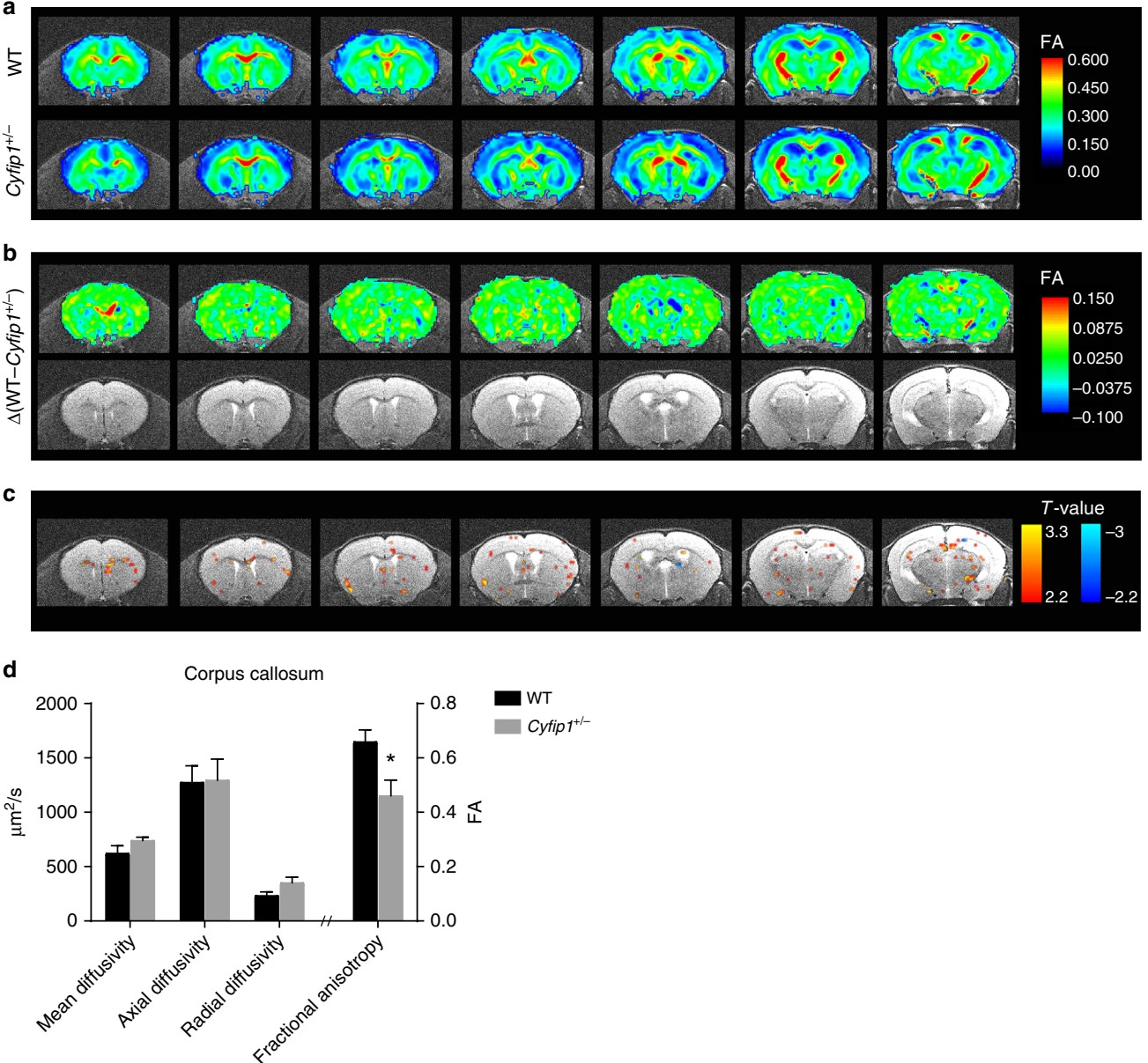

**Fig. 2** $Cyfip1^{+/-}$ mice show defects in callosal architecture. **a** Diffusion Tensor Imaging (DTI) of WT and $Cyfip1^{+/-}$ mice at postnatal day 60 (P60). Fractional anisotropy (FA) maps in which average FA values for WT and $Cyfip1^{+/-}$ mice are represented by a colour scale (WT $n = 7$ and $Cyfip1^{+/-}$ $n = 6$ mice). **b** Upper, FA differential maps between WT and $Cyfip1^{+/-}$ mice ($\Delta$(WT- $Cyfip1^{+/-}$)). Lower, representative anatomical MRI images as reference. **c** Statistical map of the FA differences between WT and $Cyfip1^{+/-}$ mice. Shown are the T-values in a colour scale ranging from 2.2 to 3.3 (equivalent to uncorrected $p$ values ranging from 0.05 to 0.005). The red scale indicates reduced FA values in the $Cyfip1^{+/-}$ mice, the blue scale increased values. **d** The graph shows mean diffusivity (MD), axial diffusivity (AD), radial diffusivity (RD) and fractional anisotropy (FA) in the corpus callosum (WT $n = 7$ and $Cyfip1^{+/-}$ $n = 6$ mice) (mean ± SEM; two-tailed $t$-test; FA *$p = 0.0165$)

difference between WT and $Cyfip1^{+/-}$ mice (Supplementary Fig. 2), suggesting that this is not likely to be the primary cause of the observed reduction in myelin thickness.

It is well-established that neuronal activity can regulate callosal myelination[52]. To investigate whether $Cyfip1$ deficiency leads to defects in neuronal activity, we measured spontaneous activity in the somatosensory cortex of acute brain slices using microelectrode arrays (MEAs) (Fig. 4a). We found that both the spike and burst rate were significantly reduced in the $Cyfip1^{+/-}$ cortices (Fig. 4b–d), indicating that there are perceptible network alterations in the adult brain. This reduction in spontaneous activity could explain the reduced myelin thickness of callosal

axons, which would in turn affect their function, potentially having repercussion on synaptic neurotransmission. To study the latter, we extracellularly stimulated the corpus callosum to elicit EPSCs in pyramidal cells of the layer II/III[53], where contralateral projections arrive through the callosal tract (Fig. 5a). Upon challenging callosal axons with trains of stimulation, slices from WT mice presented short-term synaptic depression. In contrast, in $Cyfip1^{+/-}$ slices we observed significantly reduced frequency depression together with an increased $EPSC_5/EPSC_1$ ratio (Fig. 5b, c), indicating altered presynaptic function. Altogether, this evidence corroborates the hypothesis that in $Cyfip1^{+/-}$ mice, bilateral transmission via callosal tracts is impaired.

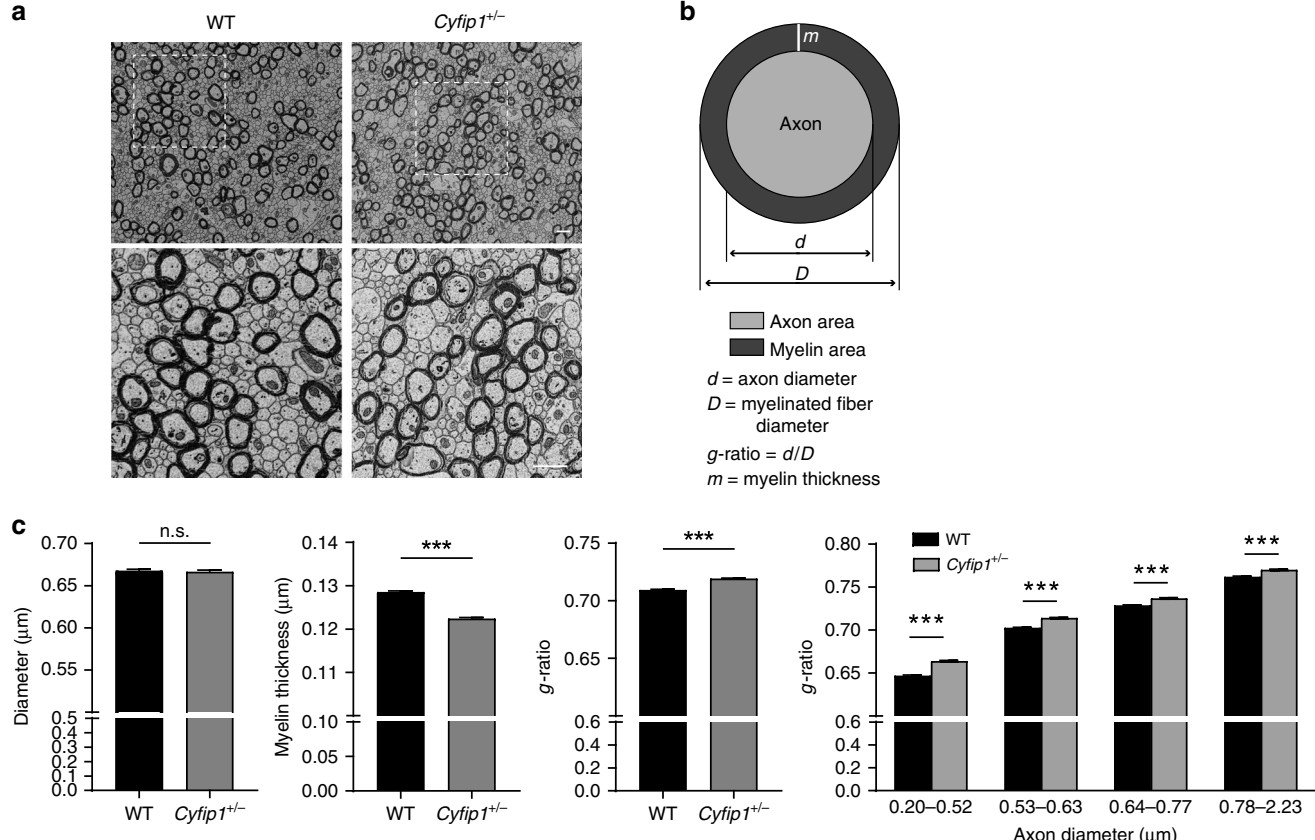

**Fig. 3** $Cyfip1^{+/-}$ mice show defects in callosal myelination. **a** Representative electron micrographs of axons in the genu of the corpus callosum (CC) in WT and $Cyfip1^{+/-}$ mice at P60. Lower panel, zoom image (scale bar 1 μm). **b** Schematic of the measured parameters. **c** Graphs show average axonal diameter, myelin thickness and g-ratio. The histogram on the right shows the g-ratio of axons with different diameters ($n > 10,000$ axons for each genotype, $n = 3$ mice for each genotype, mean ± SEM; two-tailed Mann–Whitney test; myelin thickness ***$p < 0.0001$, g-ratio ***$p < 0.0001$; Two-way ANOVA $F_{(1,20585)} = 311.8$, with Holm-Sidak's multiple comparison test, g-ratio ***$p < 0.0001$ for all axon diameters)

**$Cyfip1^{+/-}$ mice show motor coordination defects.** Callosal abnormalities have been associated with motor deficits, particularly with motor coordination, both in humans[6] and in mice[54,55]. We therefore used the accelerating rotarod and the ladder rung walking tests to investigate how the impaired bilateral connectivity and callosal transmission in $Cyfip1^{+/-}$ mice affect motor function. As shown in Fig. 6a, b, $Cyfip1^{+/-}$ mice fell off significantly earlier from the rotarod than their WT littermates, indicating reduced motor coordination. Of note, this defect was more prominent in the first trial (Fig. 6b). Interestingly, in the ladder rung walking test, $Cyfip1^{+/-}$ mice performed strikingly worse, slipping more often across trials (Fig. 6c), and in each of the three trials (Fig. 6d), than their WT littermates. To exclude that the abnormal coordination was due to reduced locomotion or muscle strength, general activity and strength were measured in the open field and hanging wire, respectively. $Cyfip1^{+/-}$ animals moved normally (Supplementary Fig. 3a–c) and had normal muscle strength (Supplementary Fig. 3d). Altogether, these findings indicate that the defects on the rotarod and the ladder rung test are indeed due to impaired motor coordination.

**$Cyfip1^{+/-}$ mice recapitulate ASD and SCZ-like behaviours.** To investigate whether this model has translational relevance, we analysed whether the $Cyfip1^{+/-}$ mice would recapitulate some ASD and SCZ-like behavioural phenotypes.

Individuals with ASD are characterized by defects in social communication and repetitive and stereotyped behaviours[56]. In addition, they present a wide range of symptoms such as defects in motor coordination, altered sensory processing, novelty seeking and decreased cognitive flexibility[57–61]. We measured sensory perception/novelty detection using the texture novel object recognition paradigm (tNORT[62]). In contrast to WT, $Cyfip1^{+/-}$ mice showed no preference for the novel vs. the familiar texture (Fig. 7a), suggesting defects in sensory processing or novelty seeking behaviour. Repetitive behaviour was assessed through scoring grooming behaviour and marble burying. In both tests $Cyfip1^{+/-}$ mice scored similarly to WT mice (Supplementary Fig. 4a, b). Finally, cognitive flexibility was assessed in two different behavioural paradigms, namely the reversal phase of Morris water maze and the visual discrimination assay using automated touchscreen-base opearant chambers. None of the two tests revealed major differences in cognitive flexibility between WT and $Cyfip1^{+/-}$ mice (Supplementary Fig. 5).

Individuals with schizophrenia present symptoms that are generally grouped in three categories: positive symptoms (e.g., hallucinations), negative symptoms (e.g., affective blunting) and cognitive impairment (e.g., working memory deficits)[63,64]. In addition, sensorimotor gating defects are one of the most robust behavioural phenotypes found in schizophrenic patients[65]. Sensorimotor gating is measured by prepulse inhibition (PPI), i.e., the attenuation of a startle reflex when preceded by a weaker prestimulation[66]. PPI is regarded as an endophenotype of SCZ and can be easily translated to preclinical models to describe SCZ-like behaviour in rodents[67]. We found that $Cyfip1^{+/-}$ mice have reduced PPI, similarly to what has been observed in patients with SCZ, and indicating decreased sensory motor gating (Fig. 7b, c).

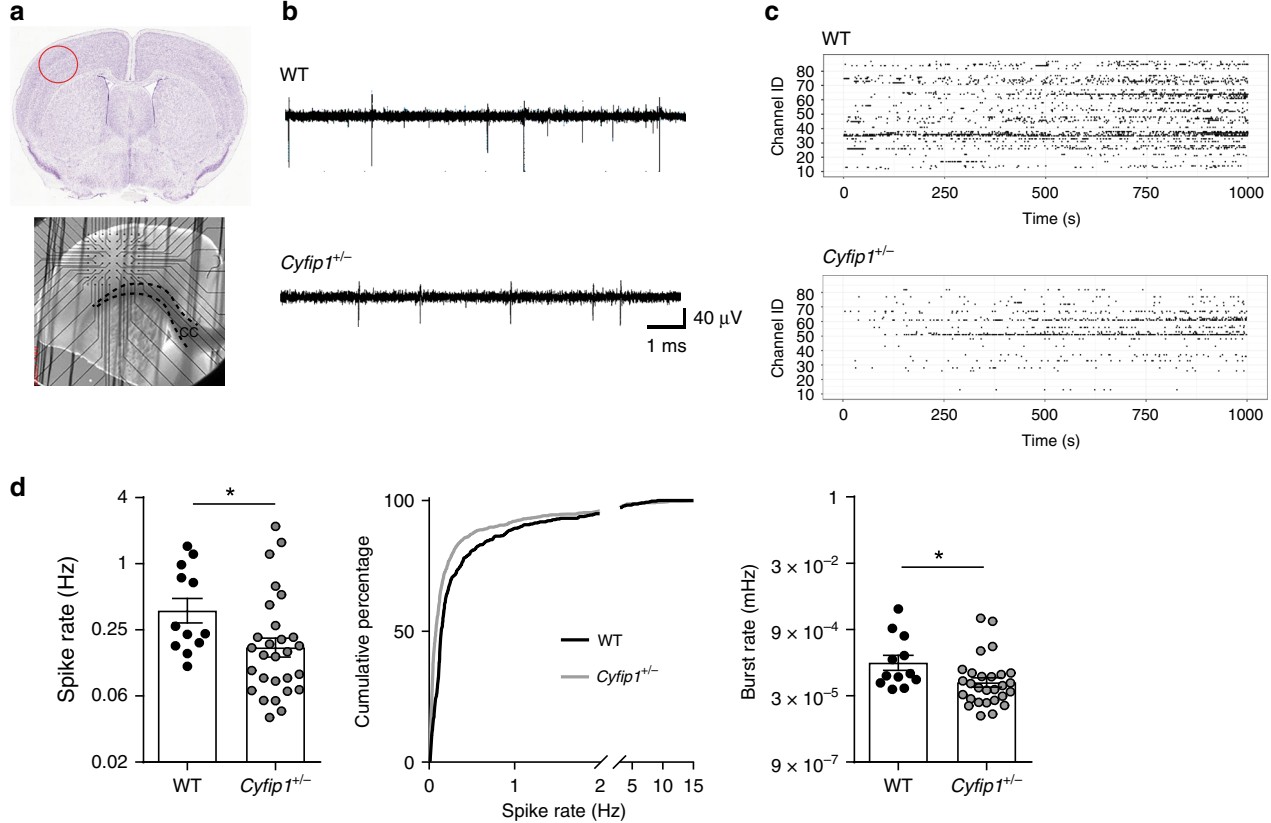

**Fig. 4** Spontaneous neuronal activity is reduced in *Cyfip1*$^{+/-}$ adult mice. **a** Upper panel, representative image of a P60 brain slice (Image credit: Allen Institute. Coronal Allen Mouse Brain Atlas (https://mouse.brain-map.org/static/atlas), with the recorded area highlighted by a red circle. Lower panel, representative image of the P60 somatosensory cortex slice positioned on the microelectrode array (MEA) system. Dashed black lines delineate the corpus callosum (CC). The field of electrodes is visible above the CC. **b** Representative traces of single electrodes recording from WT and *Cyfip1*$^{+/-}$ slices. **c** Representative raster plots of all 59 electrodes showing the spike events in WT and *Cyfip1*$^{+/-}$ cortical slice recordings. The x-axis corresponds to the recording time and the y-axis to the electrode ID. **d** Quantification of the spike rate (left) and burst rate (right, both on a logarithmic scale) in WT and *Cyfip1*$^{+/-}$ brain slices at P60 (WT n = 12 slices, 6 mice, and *Cyfip1*$^{+/-}$ n = 28 slices, 13 mice; mean ± SEM; two-tailed t-test; *p = 0.032 for spike rate and *p = 0.027 for burst rate). Centre, cumulative frequency distribution of the spike rate for WT and *Cyfip1*$^{+/-}$ mice (Kolmogorov–Smirnov test, p < 0.001)

Finally, we analysed working memory in the Y-maze, where *Cyfip1*$^{+/-}$ mice showed no deficits compared to WT (Supplementary Fig. 4c, d).

In sum, we gathered compelling electrophysiological, anatomical, functional and behavioural evidence, which all demonstrate defects in callosal connections of the adult *Cyfip1*$^{+/-}$ mouse brain. Furthermore, the *Cyfip1*$^{+/-}$ haploinsufficiency mouse model recapitulates several behavioural endophenotypes of ASD and SCZ.

## Discussion

Here, we describe an impairment in corpus callosum architecture and transmission in a mouse model of *Cyfip1* haploinsufficiency. These callosal defects have an impact on functional connectivity and behaviour that might explain the genetic association of *CYFIP1* with neuropsychiatric disorders.

We observed impaired bilateral functional connectivity (Fig. 1), which correlates with alterations in the callosal axons. Specifically we observed defects in the microstructure of the corpus callosum, determined by DTI, which correlates with reduced myelin thickness (Figs. 2 and 3), and presynaptic transmission (Fig. 5). Regarding presynaptic function, we observed that trains of stimulation lead to short-term synaptic depression in WT but not in *Cyfip1*$^{+/-}$ slices (Fig. 5). This could be caused by changes in neurotransmitter release probability[68], which would be reduced

in *Cyfip1*$^{+/-}$ slices, although we cannot exclude that other presynaptic alterations could also be the cause. Arguably, changes in synaptic transmission have an impact on spontaneous activity, as we observed (Fig. 4), and neuronal activity modulates myelination of the callosal axons[52]. In this model, the primary defect lies in the callosal axons. Alternatively, alterations in oligodendrocytes may represent the primary cause of the reduced myelination, which in turn would affect propagation of electrical signals along the callosal axons. We did not observe defects in oligodendrocyte number, suggesting that presynaptic dysfunction is likely to be fundamental in the callosal dysfunction. Interestingly, a recent in vivo study in zebrafish shows that reduced vesicle release leads to impaired myelination by reducing oligodendrocytes myelinating capacity[69]. However, further studies are needed to elucidate this cause-consequence relation and the molecular role of CYFIP1 in presynaptic function.

Even though the largest reduction in FA in the *Cyfip1*$^{+/-}$ mice was observed in the CC, other axonal tracts may also be affected. CYFIP1 is expressed across the whole brain[42,70] and therefore axonal defects in the *Cyfip1*$^{+/-}$ mice may be generalised. Indeed, both fractional anisotropy and functional connectivity seemed generally reduced across the whole brain in *Cyfip1*$^{+/-}$ mice. Similarly, defects in FC and white matter architecture have been reported across the brain in patients with ASD and SCZ, as well as 15q11.2 deletion and duplication[12,15,17,18,30], suggesting that

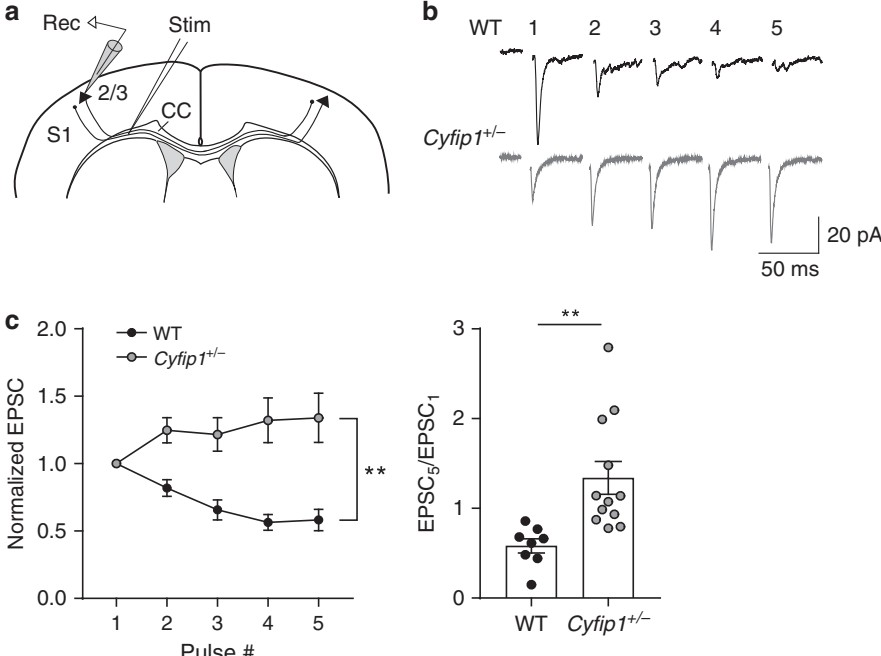

**Fig. 5** Reduced presynaptic function in *Cyfip1*[+/−] adult mice. **a** Schematic of the experimental setup. **b** Representative sample traces induced in layer II/III cortical neurons at P60 by trains of callosal stimulation (5 pulses, 20 Hz). Recordings obtained from WT (black) and *Cyfip1*[+/−] (grey) mice. **c** Left, graph of the normalised amplitude of EPSCs, and right, histogram and scatter plot of the amplitudes of the fifth EPSC, relative to the first. (WT $n = 8$ cells, three mice and *Cyfip1*[+/−] $n = 12$ cells, three mice; mean ± SEM; two-way repeated measures ANOVA, genotype main effect **$p = 0.0017$, $F_{(1,18)} = 13.53$, and interaction $p < 0.0001$; two-tailed *t*-test **$p = 0.0046$)

connectivity problems between brain areas may also contribute, in addition to the callosal defects, to neuropsychiatric disorder-related behaviours. Importantly, patients with the 15q11.2 BP1–BP2 microdeletion syndrome present thinner corpus callosum[28].

Notably, motor coordination deficits have been observed in patients with ASD[60], and a high proportion of 15q11.2 patients (42%) show motor delay[24], which may correlate with the reduced motor coordination we observed (Fig. 6). Autism spectrum disorders are a complex group of disorders characterized by social deficits, language impairment and repetitive behaviours. However, patients with ASD also present abnormal sensory processing[58], novelty seeking[61], working memory[71], cognitive flexibility[59] and sensorimotor gating[72]. Some of these phenotypes have also been associated with schizophrenia[64,65].

*Cyfip1* haploinsufficiency recapitulates several ASD and SCZ-related behavioural alterations, specifically in sensory perception/novelty seeking and sensorimotor gating (Fig. 7) as well as defective cognition and sociability[41,43]. These observations suggest that the *Cyfip1*[+/−] mouse model has face validity for ASD and SCZ.

Several other behavioural functions appear normal, such as working memory, cognitive flexibility and repetitive behaviours (this study and[41,43]) as well as general locomotion (this study and[41,43], but see also[44]). While this work was under revision[73], a study was published showing motor defects in the *Cyfip1*[+/−] mice[48]. Of interest, deficits in motor coordination, sensory perception and sensorimotor gating have been observed in other mouse models for ASD and SCZ. For instance, reduced sensorimotor gating and motor coordination have been observed in the SCZ mouse model for 22q11.2 deletion[74,75]. In addition, disruption of the ASD-associated *Shank3* gene in mice leads to reduced texture discrimination and motor coordination[62,76].

Of note, the BP1–BP2 region contains four genes; here, we show that in rodents the callosal defects and behavioural alterations are caused by the sole deletion of the *Cyfip1* gene. While we

cannot formally exclude a role for the other three genes in the region, our findings here, as well as human genetic evidence (see Introduction), indicate that human *CYFIP1* could be a causative gene in the BP1–BP2 region.

Alterations in the callosal region are not only found in patients with 15q11.2 BP1–BP2 deletions and duplications[27–30], they are also a hallmark of ASD[19] and SCZ[16]. Thus, variations in other chromosomal regions genetically associated with ASD or SCZ present similar defects in brain structure and connectivity as the *Cyfip1* haploinsufficient mice. For instance, 22q11.2 deletion carriers, a chromosomal rearrangement that is associated with SCZ, present structural abnormalities in several brain regions[77] which are recapitulated in a mouse model for the disease (Df(16) A[+/−])[78]. Similarly, studies in humans carrying the autism-associated 16p11.2 microdeletion showed reduced prefrontal connectivity and microstructural defects in the corpus callosum. These defects were also found in the mouse model for the disease, in which the callosal abnormalities were associated with increased axonal g-ratio[79]. Furthermore, *Cntnap2* KO mice, one of the first identified genes associated with ASD[80], have defects in callosal transmission and cortical myelination[53], as well as reduced functional connectivity and aberrant g-ratio of the callosal axons[81,82]. Taken together, these studies point out that neuropsychiatric disorders of different genetic origins present similar defects in brain connectivity and white matter architecture, especially in the corpus callosum, highlighting the importance of brain connectivity as a possible point of convergence for several neuropsychiatric disorders[83]. In addition, some of the areas that showed reduced functional connectivity in the *Cyfip1*[+/−] mice are part of the default mode network (DMN), a brain network that reflects functional connectivity across different brain regions during conscious-inactive tasks[84]. Although the exact function of the DMN is still unknown, DMN defects have been found in patients with SCZ and ASD and have been suggested to be key for understanding the pathophysiology of these disorders[15,85].

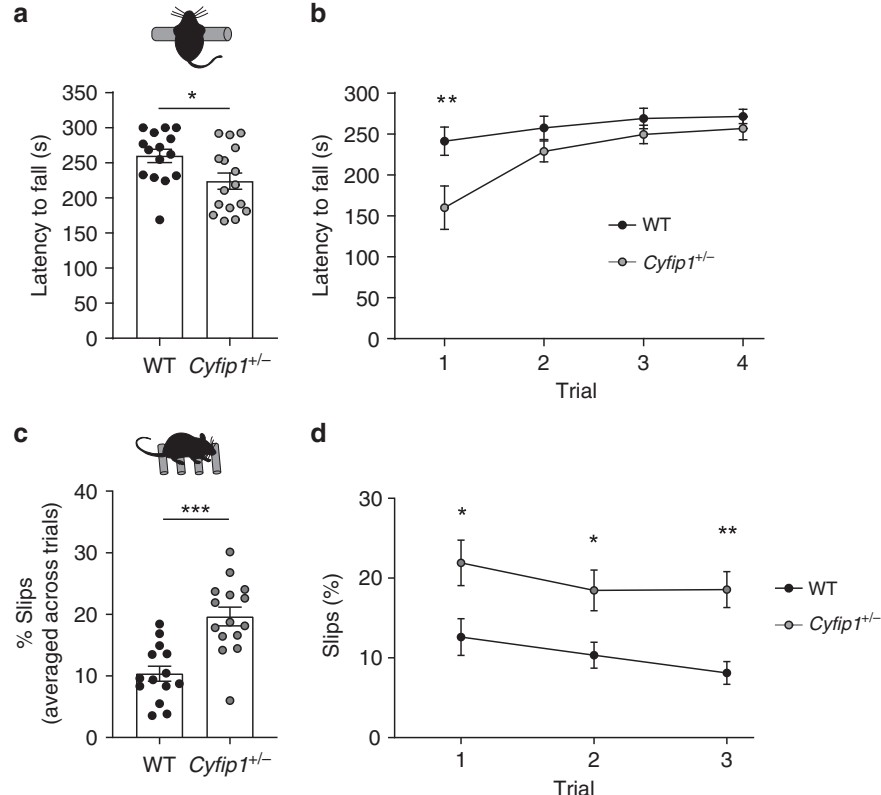

**Fig. 6** Motor coordination is reduced in Cyfip1[+/−] mice. **a** Average latency to fall measured in the accelerating rotarod across the four trials (WT $n = 15$ and Cyfip1[+/−] $n = 16$ mice; mean ± SEM; two-tailed $t$-test; *$p = 0.024$). **b** Latency to fall per trial measured in the accelerating rotarod (WT $n = 15$ and Cyfip1[+/−] $n = 16$ mice; mean ± SEM; Two-way repeated measures ANOVA; $p = 0.024$ for genotype, $F_{(1,29)} = 5.68$, trial 1 **$p = 0.0015$). **c** Percentage of slips measured in the ladder rung walking test averaged across the three trials (WT $n = 14$ and Cyfip1[+/−] $n = 15$ mice; mean ± SEM; two-tailed $t$-test; ***$p < 0.0001$). **d** Percentage of slips per trial in the ladder rung walking test (WT $n = 14$ and Cyfip1[+/−] $n = 15$ mice; mean ± SEM; Two-way repeated measures ANOVA; genotype $p < 0.0001$, $F_{(1,27)} = 22.04$, trial 1 *$p = 0.014$, trial 2 *$p = 0.038$, trial three **$p = 0.0048$ in a post-hoc Sidak test)

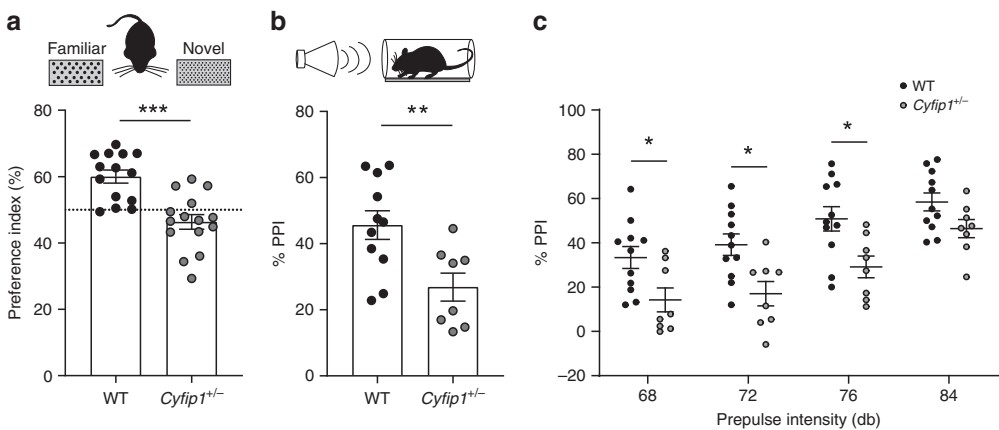

**Fig. 7** Cyfip1[+/−] mice present ASD and SCZ-related phenotypes. **a** Preference for novel vs. familiar texture (tNORT) was measured for WT and Cyfip1[+/−] mice (WT $n = 14$ and Cyfip1[+/−] $n = 15$ mice; mean ± SEM; two-tailed $t$-test; ***$p < 0.0001$). **b** Average percentage of prepulse inhibition (PPI) for all intensities is shown for WT and Cyfip1[+/−] mice (WT $n = 11$ and Cyfip1[+/−] $n = 8$ mice) (mean ± SEM; two-tailed $t$-test; **$p = 0.0081$). **c** Percentage of PPI for each prepulse intensity in WT and Cyfip1[+/−] mice (WT $n = 11$ and Cyfip1[+/−] $n = 8$ mice) (mean ± SEM; Two-way repeated measures ANOVA; genotype $p = 0.0072$, $F_{(1,17)} = 9.317$, with Sidak's multiple comparison 68 dB *$p = 0.0367$, 72 dB *$p = 0.0113$, 76 dB *$p = 0.0134$)

In conclusion, our results suggest that *CYFIP1* is most likely the gene in the 15q11.2 region which haploinsufficiency is responsible for key phenotypes of patients with ASD and SCZ. *Cyfip1* haploinsufficiency leads to alterations in anatomic and functional connectivity, thereby explaining the increased incidence of these pathologies in patients with 15q11.2 deletions. Importantly, this pathological mechanism might not be limited to

neuropsychiatric disorders caused by CNVs in the 15q11.2 region. CYFIP1 protein has been found dysregulated in SCZ independently of mutations in the *CYFIP1* locus[86], indicating that other genetic and/or environmental factors may converge on the regulation of CYFIP1 abundance or activity, rendering the protein a hub for the development of SCZ and perhaps also related disorders. Altogether, the connectivity deficits caused by CYFIP1

deficiency could therefore have a major contribution in the pathogenesis of neuropsychiatric disorders.

## Methods

**Animal care.** Animal housing and care were conducted according to the Belgian, Swiss and European laws, guidelines and policies for animal experimentation, housing and care (Belgian Royal Decree of 29 May 2013, European Directive 2010/63/EU on the protection of animals used for scientific purposes of 20 October 2010 and Swiss loi fédérale sur la protection des animaux 455). All the experimental procedures performed in Switzerland complied with the Swiss National Institutional Guidelines on Animal Experimentation and were approved by the Cantonal Veterinary Office Committee for Animal Experimentation. In all cases special attention was given to the implementation of the 3Rs, housing and environmental conditions and analgesia to improve the animals' welfare.

A 12-h light/dark cycle was used, and food and water were available *ad libitum*. The $Cyfip1^{+/-}$ mouse line, generated by gene trap at the Sanger Institute, UK, was kindly provided by Seth G.N. Grant. The gene trap cassette was inserted between exon 12 and 13. Molecular characterisation of the $Cyfip1^{+/-}$ mice was previously reported by De Rubeis et al.[38]. C57BL/6 wild type (WT) and $Cyfip1$ heterozygous ($Cyfip1^{+/-}$) male mice of 2–4 months were used. To generate the experimental animals, $Cyfip1^{+/-}$ male were crossed with WT C57BL/6 females. Animal manipulation was performed according to the ECD protocol approved by the Institutional ethical committee of the KU Leuven and Cantonal Veterinary Office Committee for Animal Experimentation.

**rsfMRI and DTI.** *Animals*: for the MRI handling procedures, adult male mice were anaesthetised with 2.5% isoflurane (IsoFlo, Abbott, Illinois, USA), which was administered in a mixture of 70% nitrogen (400 cc min$^{-1}$) and 30% oxygen (200 cc min$^{-1}$).

Resting state functional magnetic resonance imaging (rsfMRI) procedures were performed as previously described[87]. In brief, a combination of medetomidine (Domitor, Pfizer, Karlsruhe, Germany) and isoflurane was used to sedate the animals. After positioning of the animal in the scanner, medetomidine was administered subcutaneously as a bolus injection (0.3 mg kg$^{-1}$), after which the isoflurane level was immediately decreased to 1%. Five minutes before the rsfMRI acquisition, isoflurane was decreased to 0.4%. rsfMRI scans were consistently acquired 40 min after the bolus injection, during which the isoflurane level was maintained at 0.4%. After the imaging procedures, the effects of medetomidine were counteracted by subcutaneously injecting 0.1 mg kg$^{-1}$ atipamezole (Antisedan, Pfizer, Karlsruhe, Germany).

For the diffusion tensor imaging (DTI) measurements, after the handling procedures under isoflurane (2.5%), isoflurane levels were decreased to 1.5% and maintained throughout the scanning procedure. The physiological status of all animals was monitored throughout the imaging procedure. A pressure sensitive pad (MR-compatible Small Animal Monitoring and Gating system, SA Instruments, Inc.) was used to monitor breathing rate and a rectal thermistor with feedback controlled warm air circuitry (MR-compatible Small Animal Heating System, SA Instruments, Inc.) was used to maintain body temperature at (37.0 ± 0.5) °C.

*Imaging procedures*: rsfMRI procedures were performed on a 9.4T Biospec MRI system (Bruker BioSpin, Germany) with the Paravision 5.1 software (www.bruker.com). Images were acquired using a standard Bruker cross coil setup with a quadrature volume coil and a quadrature surface coil for mice. Three orthogonal multi-slice Turbo RARE T2-weighted images were acquired to render slice-positioning uniform (repetition time 2000 ms, effective echo time 33 ms, 16 slices of 0.5 mm). Field maps were acquired for each animal to assess field homogeneity, followed by local shimming, which corrects for the measured inhomogeneity in a rectangular VOI within the brain. Resting-state signals were measured using a T2*-weighted single shot EPI sequence (repetition time 2000 ms, echo time 15 ms, 16 slices of 0.5 mm, 150 repetitions). The field-of-view was (20 × 20) mm² and the matrix size (128 × 64).

DTI was performed on a 7T Pharmascan system (Bruker BioSpin, Germany). Images were acquired using a Bruker cross coil setup with a transmit quadrature volume coil and a receive-only surface array for mice. Three orthogonal multi-slice Turbo RARE T2-weighted images were acquired to render slice-positioning uniform (repetition time 2500 ms, effective echo time 33 ms, 18 slices of 0.5 mm). Field maps were acquired for each animal to assess field homogeneity, followed by local shimming, which corrects for the measured inhomogeneity in a rectangular VOI within the brain. DTI images were acquired using a multislice two-shot spin-echo EPI sequence (repetition time 5500 ms, echo time 23.23 ms, 18 slices of 0.5 mm, $b = 800$ s mm$^{-2}$, 60 DW direction). The field of view was (20 × 20) mm² and the matrix size (96 × 96).

*Image processing*: pre-processing of the rsfMRI and DTI data, including realignment, normalisation and smoothing (for the rsfMRI data), was performed using SPM8 software (Statistical Parametric Mapping, http://www.fil.ion.ucl.ac.uk). First, all images within each session were realigned to the first image. This was done using a least-squares approach and a 6-parameter (rigid body) spatial transformation. For the analyses of the rsfMRI data, motion parameters resulting from the realignment were included as covariates to correct for possible movement that occurred during the scanning procedure. Second, all datasets were normalised

to a study-specific EPI template. The normalisation steps consisted of a global 12-parameter affine transformation followed by the estimation of the nonlinear deformations. Finally, in plane smoothing was done for the rsfMRI data using a Gaussian kernel with full width at half maximum of twice the voxel size (0.31 × 0.62 mm²). All rsfMRI data were filtered between 0.01 and 0.1 Hz using the REST toolbox (REST1.7, http://resting-fmri.sourceforge.net).

Regions-of-interest (ROIs) were manually defined using the Franklin and Paxinos anatomical mouse brain atlas and the MRicron software (MRicron version 6.6, 2013, http://www.mccauslandcenter.sc.edu/mricro/): cingulate cortex, caudate putamen, hippocampus, motor cortex, orbitofrontal cortex, prelimbic cortex, retrosplenial cortex, somatosensory cortex and thalamus. The rsfMRI data were first analysed using a region-of-interest (ROI) correlation analysis, where pairwise correlation coefficients were calculated and z-transformed using an in-house programme developed in MATLAB (MATLAB R2013a, The MathWorks Inc. Natick, MA, USA). Mean z-transformed FC matrices were calculated for each group. For each brain region mean FC, i.e., mean FC of each region across the other regions in the matrix, and bilateral FC were calculated. Statistical analyses of the pairwise correlations included two-way ANOVAs to compare groups and brain regions using the Holm-Sidak correction for multiple comparisons.

Next, a seed-based analysis was performed to identify functional correlations of each region to all the voxels in the brain. Seed-based analyses were performed by first computing individual z-transformed FC-maps of the respective region using the REST toolbox, after which mean statistical FC-maps were calculated for each group in SPM8.

Each individual DTI FA-map was warped to a reference anatomical image. Then, average FA-maps for WT and $Cyfip1^{+/-}$ mice were computed and a differential map ($\Delta$(WT-$Cyfip1^{+/-}$)) was calculated. To assess statistically significant differences between the two groups, we performed a voxel-wise two-sample $t$-test.

Finally, the corpus callosum was manually delineated and DTI parameters (i.e., radial diffusivity, axial diffusivity, mean diffusivity and fractional anisotropy) were computed with MATLAB and statistical analyses were performed using two-sample $t$-tests to compare group differences.

The volume of the corpus callosum was calculated using Amira5.4 software overlaying individual CC ROIs over each subjects T2-weighted anatomical MRI images.

**Electron microscopy.** Adult mice from both genotypes were perfused, via the heart, with 100 ml of a buffered mix of 2.5% glutaraldehyde and 2.0% paraformaldehyde in 0.1 M phosphate buffer (pH 7.4), and then left for 2 h before the brain was removed, embedded in 5% agarose, and sagittal vibratome sections cut at 80 μm thickness through the midline. These sections were then post-fixed in potassium ferrocyanide (1.5%) and osmium (2%), and stained with thiocarbohydrazide (1%) followed by osmium tetroxide (2%). They were later stained overnight in uranyl acetate (1%) and washed in distilled water at 50 °C before being stained with lead aspartate at the same temperature. They were finally dehydrated in increasing concentrations of alcohol and then embedded in spurs resin and hardened at 65 °C for 24 h between glass slides. The regions containing the corpus callosum were trimmed from the rest of the section using a razor blade and glued to a blank resin block. One-micrometre-thick sections were then cut from the block face and mounted onto silicon wafers of 1 cm diameter.

The sections were imaged inside a scanning electron microscope (Zeiss Merlin, Zeiss NTS) at a voltage of 2 kV and image pixel size of 7 nm. Backscattered electrons were collected with a Gatan backscattered electron detector with a pixel dwell time of 1 μs. Multiple images of the corpus callosum were collected and tiled using the TrakEM2 plugin[88] in the FIJI software (www.fiji.sc).

A custom-made MATLAB code was used to identify and parameterise myelinated axons within the EM images. Inner (axon) and outer (axon + myelin) areas were segmented and their corresponding circular equivalent diameters calculated. The g-ratio was calculated as the ratio between the internal and external diameter (d/D) (Fig. 3b).

**Immunohistochemistry and confocal microscopy.** Adult mice from both genotypes were transcardially perfused with cold 4% PFA (pH 7.4). Coronal sections were obtained using a cryostat (Leica CM 3050 S; 20 μm thick). Free-floating sections from the anterior part of the CC (bregma 1.1 mm to 0.74 mm) were permeabilized and blocked at RT in phosphate-buffered saline (PBS, pH = 7.4), 0.3% Triton X-100, 0.5% BSA and 2% goat normal serum. Slices were incubated overnight at 4 °C with the primary antibodies (rabbit anti-Olig2 1:10000, AB9610 Millipore; mouse anti-CC-1 1:200, Sigma OP80) and 2 h at RT with the secondary Alexa-conjugated antibodies (Life Technologies). DAPI was used for nuclei visualisation. Brain sections were mounted using Mowiol (Sigma).

Confocal images were acquired on a Leica SP8 microscope. Mosaic stack images were taken covering the corpus callosum around the midline. The number of Olig2$^+$ and CC1$^+$ cell was manually assessed using the Image J software.

**Microelectrode array recordings.** Cortical slices were prepared from adult male mice to study the effects of $Cyfip1$ haploinsufficiency in spontaneous activity. Briefly, the mice were anaesthetised with isoflurane and the brains were quickly

removed and placed in bubbled ice-cold 95% $O_2$/5% $CO_2$-equilibrated solution containing (in mM): NaCl 125, KCl 2.5, $NaH_2PO_4$ 1, $MgCl_2$ 1.2, $CaCl_2$ 0.6, Glucose 11, $NaHCO_3$ 26 (95% $O_2$, 5% $CO_2$). Coronal brain slices (350 μm thick) were made using a vibratome (Leica Biosystem). Slices were kept for 30 minutes at 33 °C and for 1 hour at RT in artificial cerebrospinal fluid (ACSF) containing (in mM): NaCl 125, KCl 2.5, $NaH_2PO_4$ 1, $MgCl_2$ 2, $CaCl_2$ 1, Glucose 11, $NaHCO_3$ 26 (95% $O_2$, 5% $CO_2$).

A single slice was placed in a multi electrode array (MEA) chamber (Multichannel systems, Reutlingen, Germany) positioning the electrode array on the somatosensory cortex. Each MEA is composed of 59 TiN electrodes organised in an 8 × 8 grid layout. The electrode distance is 200 μm, and the electrode diameter is 30 μm. The slice was stabilised in the MEA chamber using a platinum anchor and was continuously perfused with modified ACSF containing (in mM): $MgCl_2$ 0.75, $CaCl_2$ 1.4, (95% $O_2$, 5% $CO_2$). Neuronal activity, sampled at 10 kHz and filtered at 100 Hz, was recorded with a MEA2100-acquisition System (Multichannel systems, Reutlingen, Germany). The data were acquired with the MultiChannel Experimenter 2.8 and extracted for analysis using the MultiChannel Analyzer 2.6 (Multichannel systems, Reutlingen, Germany). For spike detection, the threshold was set at −5.5 s.d. of the noise amplitude. An event was considered a burst if it was composed of at least five consecutive spikes with an inter-spike interval of <50 ms. Data formatting, detection of bursts and events as well as the statistical analyses were performed with a custom-written script in R (software version 3.4.3).

**Patch clamp recordings.** Animals were decapitated and the brain was quickly extracted. Slicing was performed in bubbled ice-cold 95% $O_2$/5% $CO_2$-equilibrated solution containing (in mM): choline chloride 110; glucose 25; $NaHCO_3$ 25; $MgCl_2$ 7; ascorbic acid 11.6; sodium pyruvate 3.1; KCl 2.5; $NaH_2PO_4$ 1.25; $CaCl_2$ 0.5. Coronal slices (250 μm) were stored at ~ 22 °C in 95% $O_2$/5% $CO_2$-equilibrated artificial cerebrospinal fluid (ACSF) containing (in mM): NaCl 124; $NaHCO_3$ 26.2; glucose 11; KCl 2.5; $CaCl_2$ 2.5; $MgCl_2$ 1.3; $NaH_2PO_4$ 1. Recordings (flow rate of 2.5 ml min$^{-1}$) were made under an Olympus-BX51 microscope (Olympus) at 32 °C. Currents were amplified, filtered at 5 kHz and digitised at 20 kHz. Access resistance was monitored by a step of −4 mV (0.1 Hz). Experiments were discarded if the access resistance increased more than 20%. A bipolar stimulating electrode was placed in the corpus callosum, and EPSCs were evoked in layer II/III pyramidal neurons while holding the cells at −60 mV. Five pulses at 20 Hz were delivered at 0.1 Hz to measure paired pulse ratios. The internal solution contained (in mM): $CH_3CsO_3S$ 120; CsCl 10; HEPES 10; EGTA 10; sodium creatine-phosphate 5; $Na_2ATP$ 4; $Na_3GTP$ 0.4; pH 7.3

**Behavioural assays.** *Open field*: exploration was observed in an open field setup with a square surface area of 45 cm². Mice were placed in the open field for 10 min and movements were recorded with EthoVision XT 14 tracking software (Noldus, Wageningen, The Netherlands). A central zone was defined as a square surface area 15 cm equidistantly from all 4 walls of the open field. Total distance travelled as well as centre entries were measured.

*Rotarod*: motor coordination was tested on the accelerating rotarod as previously described[89]. Briefly, mice were first trained for 2 min on the rotarod at constant speed (4 rpm). They were subsequently tested on four 5-min trials interleaved with 10 min rest. During the test trials, mice were placed on a rotating rod that accelerated from 4 to 40 rpm in 5 min and the latency to fall off the rod was recorded.

*Self-grooming*: animals were placed individually in a transparent cylinder (7 cm diameter, 20 cm high), and self-grooming activity was monitored for 10 min with a frontally placed camera. Time spent grooming was manually scored with EthoVision XT 14 software (Noldus, Wageningen).

*Marble burying task*: mice were placed individually in a cage of 32 × 26 cm with a bedding height of 5 cm, and 20 marbles, positioned in 5 rows of 4 marbles. The number of unburied marbles was assessed after 30 min, and the percentage of buried marbles (≥2/3 covered) calculated.

*Prepulse inhibition (PPI)*: the prepulse inhibition test was adapted from a previously published protocol[90]. Briefly, mice were placed individually into the chamber (Med Associates, Inc., St. Albans, VT) and exposed to an acclimation period (5 min + white noise at 65 dB) and a three-block session. Block 1 and 3 were composed of five pulse-alone (white noise, 120 dB, 40 ms) trials each. Block 2 was composed of 5 prepulse-pulse trials with a variable intertrial interval of 15 s (ranging from 12 to 30 s). The prepulse had variable intensities of 0, 3, 7, 11 and 19 dB above the background (white noise: 65, 68, 72, 76, 84 dB) and were followed (80 ms onset-onset delay) by a pulse sound (white noise, 120 dB, 40 ms). Each prepulse-pulse trial was presented 10 times in a pseudo-randomised sequence. The PPI % was calculated as follows: % PPI = 100 − {[(startle response for prepulse + pulse)/(startle response for pulse-alone)] x 100}, where the startle response for pulse-alone was measured as the average of pulse-alone trials of block 2.

*Morris water maze (MWM)*: to assess cognitive performance and flexibility in mice, we performed a spatial learning and reversal protocol in the Morris water maze as previously described[91]. Briefly, a large circular black pool (142 cm diameter) was filled with opacified water (26 °C) to a depth of 20 cm. A circular escape platform (12 cm diameter) was hidden 1 cm below the water surface at a fixed position. The circular pool was situated in the centre of the test room (2.9 m x

2 m) with prominent visual cues hanging on all four room corners. Mice were trained to find the hidden platform during two sets of 5-day trainings (two resting days between each set). On each training day, 4 trials were performed with an intertrial interval of 30 min. Mice were placed into the pool at one of 4 pseudo-randomly chosen locations at the pool border. Mice were guided to the platform when they failed to find the platform within 2 min. After 15 s on the platform, they were returned to their cages underneath a heating lamp. Retention memory was evaluated on probe trials presented two days after the last day of each set, where the platform was removed and swimming path was recorded for 100 s. Time spent in each quadrant was measured. After the second probe trial, we moved the platform position to the opposite quadrant and trained the mice for an additional 4 days. After the fourth day of reversal training, mice were subjected to a final probe trial. Distance travelled as well as time spent in each quadrant were recorded with Ethovision XT 14 (Noldus, Wageningen).

*Y-maze arm alternation*: forced arm alternation in the Y-maze is a test to measure the propensity of rodents to explore novel environments[92]. Visual cues were placed on the room walls. The three arms (36 cm long x 6 cm wide x 13 cm high, Ugo Basile, Italy) were separated by 120°. Mice were tested in two phases. In the first phase, one arm was blocked by an opaque removable wall and mice placed at the end of one arm (i.e. start arm) were allowed to explore freely for 5 min the start and the open arm. The mouse was afterwards returned to the home cage and remained there for 1 min. In the second phase, all arms were accessible and mice could freely explore all three arms for 2 min. The amount of time visiting the previously unvisited arm as well as the familiar arm was recorded with EthoVision XT 14 software (Noldus, Wageningen). A preference index was calculated based on [Time$_{novel}$/(Time$_{novel}$ + Time$_{familiar}$)] × 100. The position of the novel arm was counterbalanced between animals and the apparatus was cleaned with 5% ethanol/water solution between each animal.

The spontaneous alternations in the Y-maze measures short-term spatial novelty detection and is considered a test for working memory[93]. Mice were placed in the middle of the maze and were allowed to freely explore all three arms for 5 min. The number of arm entries, as well as the number of alternations was recorded. An arm entry was considered when all four limbs entered the arm. An alternation is considered when, in a moving triplet of arm visits, all three arms have been visited (e.g. 1-2-3, 3-1-2, 2-1-3, etc). The alternation score was calculated as 100×(number of alternations/number of visit triplets).

*Texture novel object recognition task (tNORT)*: an adaptation of the novel object recognition task was used as previously described[62,94]. Briefly, during the first two days of the experiment, mice were placed for 10-min inside an open field arena with opaque walls (45 × 45 × 45 cm) to ensure habituation to the environment before the test. The bottom of the arena was covered with bedding material. Mouse activity was tracked with EthoVision XT 14 software (Noldus, Wageningen, The Netherlands). Distance travelled was measured. On the third day, a texture-based novel object recognition task was introduced. During the first phase, the mouse could freely explore for 10 min two objects that were identical in texture and appearance (Familiar; two 50 ml Falcon tubes wrapped with gradient 80 sandpaper). After 5 min retention period, the mouse was reintroduced in the arena. During the second phase, one of the objects was replaced by a visually identical but texturally different object (Novel; 50 ml Falcon tube wrapped in gradient 100 sandpaper) and mouse explorative behaviour towards the two objects (at a distance of 5 cm) was recorded. Exploratory behaviour was defined as close proximity towards the object (nose orientated towards the object at a body centre distance of <5 cm). The location of the novel object was counterbalanced between animals. The novelty preference index was calculated as [Time$_{novel}$/(Time$_{familiar}$ + Time$_{novel}$)] × 100.

*Hanging wire*: the test is adapted from Aartsma-Rus et al.[95]. In brief, mice, held at the base of the tail, were guided to grasp the midpoint of a wire (length: 60 cm, height: 30 cm, thickness: 3 mm) with both forepaws and then released. The average fall latency of each mouse was recorded. The test was repeated three times with a 30-s intertrial interval and a cut-off of 5 min was used.

*Ladder rung walking test*: the ladder rung walking test was adapted from previously described[96]. Briefly, the setup consisted of two transparent sidewalls and metal rungs (2.7 mm diameter) that were inserted in the sidewalls to create a horizontal ladder with a regular distance of 1 cm between rungs (regular interval variant). The two sidewalls were separated by 3 cm, creating a narrow corridor of 32 cm long and 10 cm high for the mouse to traverse through towards a home cage at the end of the corridor. The ladder was elevated from the table by 20 cm.

Mice were trained to cross the ladder within 2 min from the start of the corridor to the safe home cage. Mice were motivated to cross the ladder and did not require further reinforcement. First, mice were allowed to freely explore the apparatus and walk along the corridor for 2 min. Afterwards, all animals were tested for 3 trials with intertrial interval of 15 min.

A 17-cm-long section of the corridor was recorded with a horizontally placed camera and videos were analysed afterwards. The number of foot slips from both fore and hind paws and the total number of steps taken to traverse the corridor were measured. The proportion of foot slips per animal per trial is reported. Foot slips were defined as a completely missing or falling off a rung while walking across the ladder.

*Touchscreen*: visual discrimination and reversal learning were tested using instrumental conditioning procedure using the touchscreens (Campden Instruments LTD, Loughborough, Leics., UK) as previously described[97]. In brief,

mice were trained to discriminate two visual images presented at the same time. Correct choices were rewarded with strawberry milkshake, while incorrect choices were penalised with 5 s timeout (indicated by chamber lights switching on). After training the animals to select and touch a visual image during a rigorous shaping procedure, animals were presented 2 visual stimuli (pairwise discrimination of correct and incorrect stimulus) that they needed to recognise and select to obtain a reward. After reaching criterion (80% correct trials on three consecutive days), the rule was reversed and the formerly incorrect stimulus was rewarded (reversal phase). Each session was limited to maximally 30 correct trial or 60 min. Number of trials to criterion was assessed for initial discrimination and reversal learning. Responses were recorded using the Abet II software.

**Statistics**. Statistical analyses were performed with SPSS and MATLAB for the MRI and DTI experiments, and with GraphPad Prism or R for EM, electrophysiology, and behavioural experiments. The statistical tests used are listed in the respective Figure legends. In brief, unpaired two-tailed Student's $t$-tests or non-parametric two-tailed Mann–Whitney $U$-tests were used for comparisons between two groups. Multiple-$t$-test with Holm-Sidak multiple comparison correction was used for the rsfMRI experiments. For cumulative frequency distribution, Kolmogorov-Smirnov test was used. For comparisons of more than two groups Two-way ANOVA or Two-way Repeated Measures ANOVA with Holm-Sidak's multiple comparison test was used. For all analyses, $P$-values < 0.05 were considered significant and annotated as follows: *$p < 0.05$, **$p < 0.01$, ***$p < 0.001$. Results were presented as mean ± standard error of the mean (SEM).

**Reporting summary**. Further information on research design is available in the Nature Research Reporting Summary linked to this article.

## Data availability
All relevant data are available from the authors upon request.

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

## Acknowledgements

We thank Joanna Viguie, Karin Jonckers and Jonathan Royaert for valuable technical assistance and Annick Crevoisier and Eef Lemmens for excellent administrative support. We are grateful to Annette Gärtner, Eleonora Rosina, Laura D'Andrea, Esperanza Fernández, Vittoria Mariano, and all the members of the Bagni laboratory for helping during the development of this project with scientific discussions. We would like to thank Graham Knott and Gadea Mata for their assistance in imaging and image processing of EM and immunohistochemistry data, Wim Vanduffel (KU Leuven) for supporting Marcelo Armendáriz and Egbert Welker for scientific discussions. We are grateful to Muna Hilal for helping in setting up the MEA system and to Leonardo Restivo (NEUROBAU) for advice on behavioural assays. We also would like to thank Romano Regazzi, Beat M. Riederer and Matthijs Verhage for providing us antibodies during the course of the study. This work was supported by KU Leuven funds (OTF), FWO G088415N,

SNSF NCCR Synapsy 51NF40-158776, SNSF 310030-182651, Novartis Foundation for Medical-Biological Research and Canton Etat de Vaud. N.D.I. was recipient of an FWO aspirant fellowship. D.S. is recipient of an FWO (12R1119N) and IWT (13160) fellowships.

## Author contributions

Conceptualisation by N.D.I. and C.B.; N.D.I. participated in all the experiments described in this work. MRI imaging by D.S. and A.V.d.L., electrophysiology by A.V., M.T. and M.M, Custom-written R scripts for MEA and behavioural experiments by A.C.L., Custom-made MATLAB code for EM analysis and DTI maps by M.A.; V.M., D.G., R.D., T.A., K.W.L. and A.B.S helped with data analysis and manuscript preparation. A.R.S. and Z.C.V. helped with the behavioural experiments.

## Additional information

**Competing interests:** The authors declare no competing interests.

