## [Peer Review File · Nature Communications]

Reviewers' comments:

Reviewer #1 (Remarks to the Author):

In this study by Claudia Bagni and colleagues, the authors investigate whether heterozygous loss of the autism-associated protein CYFIP1 leads to alterations in brain connectivity in mice. CYFIP1 is one of the genes affected by a CNV in the 15q11.2 region, which is associated with autism and schizophrenia and which is known to cause abnormalities in white matter volume and functional connectivity. In this study, the authors show that heterozygous deletion of CYFIP1 in mice leads to a reduction in functional connectivity as assessed by resting state fMRI, and that this is accompanied by a decrease in myelin thickness in callosal fibers and a change in callosal presynaptic function. Moreover, the authors report an impairment in motor coordination in these mice. Together, these data show that deletion of CYFIP1 may contribute to the phenotypes observed in patients with CNVs in the 15q11.2 locus.

Overall, the study appears thoroughly conducted, the manuscript is well written (albeit very concise, at the expense of some relevant details) and the results will be of interest to the autism research community. However, the study is purely descriptive, with very little mechanistic insight into how any of the observed abnormalities are caused or how they are related to each other, which limits the usefulness of the current data set. Moreover, a number of related phenotypes have already been reported in CYFIP1 heterozygous mice, including changes in presynaptic function and locomotor effects in an open field test (Chung et al. 2015, Brain Res. 1629: 340-350, which appears to not be cited in the manuscript). In its current form, without substantial additional mechanistic experiments, this manuscript may therefore be more suitable for a more specialized journal.

Specific comments:

1. What are the mechanisms by which *Cyfp1* deficiency results in reduced myelination and reduced presynaptic function? The authors briefly speculate in the discussion that reduced myelination may either be a primary effect or a secondary consequence of changes in synaptic transmission, but they offer no experimental evidence for either mechanism. Moreover, there is no mechanistic insight or even discussion on how the changes in presynaptic function may occur.
2. Can any of the observed phenotypes be normalized or exacerbated by manipulations that alter myelination? This might indicate that the reduced myelination plays a causal role in these phenotypes rather than being an unrelated consequence of the known alterations in synaptic transmission.
3. While resting state fMRI is a useful basic measure of connectivity, it provides no insights into the functional connectivity that determines information processing during behavioral performance. A more relevant measure for understanding how *Cyfp1* deficiency affects information processing would be to investigate functional connectivity during certain behaviors e.g. using in vivo electrophysiology.
4. The authors show that *Cyfp1* +/- mice show reduced motor coordination on an accelerating rotarod, which they use as a behavioral measure of impaired callosal transmission. However, changes in motor coordination can result from a large number of changes in the brain that are unrelated to callosal function. In order to link this behavioral impairment to callosal function, it would be necessary to delete CYFIP1 specifically from the callosum and repeat the rotarod experiments.

5. The authors conclude that “Cyfip1 haploinsufficiency is directly responsible for key phenotypes of patients with ASD and SCZ, thereby explaining the increased incidence of these pathologies in patients with 15q11.2 deletions” (p9). However, there are three other genes that are deleted in these patients, and the present study cannot rule out that any of these other genes may contribute equally or even to a greater extent to the observed phenotypes. The authors should include a discussion of the role of these other genes and their reason for ruling out a contribution from these genes.

Reviewer #2 (Remarks to the Author):

Dominguez-Iturza et al.

This manuscript is an extremely well thought out and nicely put together paper on the autism/schizophrenia associated gene CYFIP1. The authors have examined a mouse model with a heterozygous mutation of Cyfip1. They have assessed the mouse model using resting-state fMRI, DTI, which they have confirmed their results with EM, Behavioural assessments and Electrophysiology. The report is in-depth and well described. I have some minor issues, clarifications and suggestions, however.

- 1) I need a better explanation of the ROIs used in both the rs-fMRI as well as the DTI. The regions are not shown, just mentioned in the text. I would like to see the regions of interest in a map and a better explanation of how they were delineated.
 - a. Was the corpus callosum region used for the DTI measurements manually segmented? I would also like to have more information about that region and how it was segmented
- 2) Differences in FA between groups are displayed in Figure 2b. Why did you not just look for significant differences voxelwise? A difference map is somewhat less sophisticated and gives little to no information about significance.
 - a. Were there any tests used to account for multiple comparisons in the DTI examination? Or was the difference map used for directing the findings, so it was deemed unnecessary? An FA difference t-test p-value of 0.016 may not hold up to multiple comparisons. I'm assuming more was done as it was mentioned in the results that FA was globally decreased in white matter structures outside of the fimbria where there was an increase. If these comments are mentioned off of the difference map then they are not significant and then should not be stated as fact.
- 3) No diffusion differences were detected (MD, AD, RD), but yet there was an FA difference. In looking at figure 2c it is likely a small (insignificant increase in RD) that is leading to the drop in FA, consistent with the myelination differences found. This should be discussed.
- 4) More information about the sequence parameters are needed for both MRI sequences, especially the DTI sequence. The b-value of 800 s/mm² is also extremely low. Why was this chosen? Gradient constraints?
- 5) With all the delineated regions of interest and the registration procedure already done using deformation based morphometry. Why are there no volume measurements done? Was the corpus callosum smaller as well?

Reviewer #3 (Remarks to the Author):

Manuscript Number: Nature Communications manuscript 18-35561

Title: The autism and schizophrenia-associated protein CYFIP1 regulates bilateral brain connectivity

Bagni and colleagues have assessed brain connectivity by numerous imaging and

electrophysiological metrics using a model of dysregulated *Cyfp1*. The authors present a strong case for *Cyfp1*'s role in numerous structural abnormalities. Secondary, they present a small component that suggests these *Cyfp* alterations in structures and physiology are key in behavioral and functional outcomes by 2 behavioral assays important for motoric functional phenotyping.

- This sentence is awkward and incorrect as stated "...as the gene most likely causally associated with SCZ and ASD..." there is not a single gene that causes either disease and certainly associations are not causal.
- *Cyfp1* only receives a score of 3 by SFARI genome browser, meaning some suggestive evidence, but not very high so the importance to ASD was overstated.
- The illustration in 6A is unessential.
- The two assays shown and tested are the bare minimum of behavioral testing for MOTOR dysfunction and lack of motor coordination. They tell you no other information. Since only simple motor outcomes can be concluded from these two assays, it was unimpressive for a "functional" connection.
- It is unclear why, given that this model is a theoretical model of SCZ and ASD, no behavioral testing was performed that has translational relevance such as working memory for SCZ and/or social or lack of cognitive flexibility for ASD
- The methods of running the rotarod are not standard since 10 minute resting intervals does not seem like substantial rest period.
- There only seems to be significant difference at trial 1 and timepoint 1 which MEANS there is no difference in motor coordination. What is averaged in Panel 6B to detect this difference? This is not standard reporting and only further convinced me that there were not genotype differences in the behaviors tested.
- For open field, I would not use distance moved but distance traveled, and horizontal and/or vertical activity counts, however there are clearly NO genotype differences. BUT 10 minutes in the open field is not standard and extremely short. Most behavioral cores use 30 minutes to 1 hour sessions to make determinations on motor differences.
- To try and make conclusions about a functional role in disorders of the brain and behavior by rotarod and 1 open field metric is too far of stretch.
- It is more likely that this effect was not present to begin with rather than that this small defect was ameliorated after repeated trials in rotarod. To exclude that the abnormal coordination was due to reduced locomotion.
- No core domains of the ASD phenotype were even tested.
- There is no analysis of repetitive behaviors and no analysis of instance on sameness or preservative learning. Nor have they shown reciprocity in social behaviors like in a dyad social interaction.

Dominguez et al.

Reviewers' comments:

Reviewer #1 (Remarks to the Author):

In this study by Claudia Bagni and colleagues, the authors investigate whether heterozygous loss of the autism-associated protein CYFIP1 leads to alterations in brain connectivity in mice. CYFIP1 is one of the genes affected by a CNV in the 15q11.2 region, which is associated with autism and schizophrenia and which is known to cause abnormalities in white matter volume and functional connectivity. In this study, the authors show that heterozygous deletion of CYFIP1 in mice leads to a reduction in functional connectivity as assessed by resting state fMRI, and that this is accompanied by a decrease in myelin thickness in callosal fibers and a change in callosal presynaptic function. Moreover, the authors report an impairment in motor coordination in these mice. Together, these data show that deletion of CYFIP1 may contribute to the phenotypes observed in patients with CNVs in the 15q11.2 locus.

Overall, the study appears thoroughly conducted, the manuscript is well written (albeit very concise, at the expense of some relevant details) and the results will be of interest to the autism research community. However, the study is purely descriptive, with very little mechanistic insight into how any of the observed abnormalities are caused or how they are related to each other, which limits the usefulness of the current data set. Moreover, a number of related phenotypes have already been reported in CYFIP1 heterozygous mice, including changes in presynaptic function and locomotor effects in an open field test (Chung et al. 2015, Brain Res. 1629:340-350, which appears to not be cited in the manuscript). In its current form, without substantial additional mechanistic experiments, this manuscript may therefore be more suitable for a more specialized journal.

We thank the reviewer for the helpful comments. We have revised the manuscript providing additional data and further elaborated our results. Specifically we have extensively characterized the behaviour of the CYFIP1 heterozygous mice, providing new insights into the effect of CYFIP1 deficiencies on ASD and SCZ – like behaviours.

We apologize for the missing reference- Chung et al. 2015,- which has been appropriately cited in the revised version.

Specific comments:

1. What are the mechanisms by which *Cyfp1* deficiency results in reduced myelination and reduced presynaptic function? The authors briefly speculate in the discussion that reduced myelination may either be a primary effect or a secondary consequence of changes in synaptic transmission, but they offer no experimental evidence for either mechanism. Moreover, there is no mechanistic insight or even discussion on how the changes in presynaptic function may occur.

The reviewer raises a very important point. We have performed several experiments to gain more insights into the role of CYFIP1 in myelination and presynaptic function.

- 1) We have performed immunohistochemistry in the corpus callosum of adult WT and *Cyfp1*^{+/-} mice and analysed the number of immature (Olig2⁺) and mature (CC1⁺) oligodendrocytes (Supplementary Figure 2). This number does not change between WT and *Cyfp1*^{+/-} mice, suggesting that CYFIP1 does not regulate oligodendrocyte proliferation or maturation, and that the reduced myelin thickness observed could be due to defects in presynaptic functions.

- 2) We have analysed a few presynaptic proteins at the level of expression and observed no differences between WT and *Cytip1*^{+/-} mice (Figure 1 for the reviewers). We believe that CYFIP1 may play a crucial role in the regulation of presynaptic function, but to understand whether this regulation is at the level of presynaptic release, presynaptic calcium sensors, vesicle number or neurotransmitter abundance a much more thorough study on this topic is required that we plan to address in a follow up project.
- 3) We have further discussed how the interplay myelin defects-presynaptic changes may underlie the callosal defects.

Figure 1 for the reviewer. Abundance of presynaptic proteins in the cortex of WT and *Cytip1*^{+/-} mice (n = 7 WT and 7 *Cytip1*^{+/-}; t-test with Holm Sidak multiple comparison).

2. Can any of the observed phenotypes be normalized or exacerbated by manipulations that alter myelination? This might indicate that the reduced myelination plays a causal role in these phenotypes rather than being an unrelated consequence of the known alterations in synaptic transmission.

As mentioned above, the reduced myelin thickness is most probably due to the reduced neuronal activity, which negatively regulates axonal myelination. The suggested experiment is technically very challenging. To our knowledge, a way to manipulate myelin directly is through the use of cuprizone (Torkildsen et al 2008) generally used in the MS field. We consider this model too aggressive and not suitable for our research on neuropsychiatric disorders.

3. While resting state fMRI is a useful basic measure of connectivity, it provides no insights into the functional connectivity that determines information processing during behavioral performance. A more relevant measure for understanding how *Cyfp1* deficiency affects information processing would be to investigate functional connectivity during certain behaviors e.g. using in vivo electrophysiology.

We appreciate the reviewer comment and understand the concern; however, MRI is a very powerful tool for being a non-invasive translational technology. This approach allowed us to compare MRI data obtained in patients with the *Cyfp1*+/- mice suggesting that this is indeed a face validity model for ASD and SCZ-like phenotypes.

Performing in vivo electrophysiology during specific behaviours would indeed provide more insights into the role of CYFIP1 in neuronal activity during active tasks, however, these experiments will require significantly more work and would be out of the scope of this paper. Nevertheless, this is a very interesting question that we plan to address in a follow up project.

4. The authors show that *Cyfp1* +/- mice show reduced motor coordination on an accelerating rotarod, which they use as a behavioral measure of impaired callosal transmission. However, changes in motor coordination can result from a large number of changes in the brain that are unrelated to callosal function.

We agree with the reviewer. The corpus callosum is not the only brain region involved in motor coordination. For instance, the motor cortex, brain region that we also found significantly affected in bilateral connectivity, also plays a crucial role. We have selected motor coordination also inspired by the work of (Yoo et al 2016, Franco-Pons et al 2007) on mouse models with callosal defects and reduced motor coordination in the rotarod test.

In order to link this behavioral impairment to callosal function, it would be necessary to delete CYFIP1 specifically from the callosum and repeat the rotarod experiments

Concerning the reviewer suggestion, we are not aware of any methodology to silence CYFIP1 specifically from the corpus callosum. To our knowledge, every possibility would require silencing of CYFIP1 from the layer 2/3 pyramidal neurons in the motor and somatosensory cortex (which project their axons through the corpus callosum) and therefore it would remain difficult to discriminate if the defect comes from the corpus callosum or from the neurons silenced in the cortical regions.

5. The authors conclude that “*Cyfp1* haploinsufficiency is directly responsible for key phenotypes of patients with ASD and SCZ, thereby explaining the increased incidence of these pathologies in patients with 15q11.2 deletions” (p9). However, there are three other genes that are deleted in these patients, and the present study cannot rule out that any of these other genes may contribute equally or even to a greater extent to the observed phenotypes. The authors should include a discussion of the role of these other genes and their reason for ruling out a contribution from these genes.

We understand the referee point and we have now changed this sentence in the revised version. In addition, it is worth mentioning that in the SFARI database CYFIP1 is in a category with a higher score compared to the other 3 genes.

Reviewer #2 (Remarks to the Author):

Dominguez-Iturza et al.

This manuscript is an extremely well thought out and nicely put together paper on the autism/schizophrenia associated gene CYFIP1. The authors have examined a mouse model with a

heterozygous mutation of *Cyfp1*. They have assessed the mouse model using resting-state fMRI, DTI, which they have confirmed their results with EM, Behavioural assessments and Electrophysiology. The report is in-depth and well described. I have some minor issues, clarifications and suggestions, however. We thank the reviewer for the helpful comments and constructive feedback that helped us to improve the manuscript.

1) I need a better explanation of the Region Of Interest used in both the rs-fMRI as well as the DTI. The regions are not shown, just mentioned in the text. I would like to see the regions of interest in a map and a better explanation of how they were delineated.

Following the reviewer suggestion we have now specified in the Material and Methods that the rsfMRI regions of interest were manually defined using the Franklin and Paxinos anatomical mouse brain atlas and the MRicron software (MRicron version 6.6, 2013, <http://www.mccauslandcenter.sc.edu/mricron/>). In addition we have now included the maps with the defined ROIs (Supplementary Figure 1a).

a. Was the corpus callosum region used for the DTI measurements manually segmented? I would also like to have more information about that region and how it was segmented

The reviewer is correct. The corpus callosum was manually delineated. We have now included the maps with the delineated ROI (Supplementary Figure 1b) and revised the text accordingly.

2) Differences in FA between groups are displayed in Figure 2b. Why did you not just look for significant differences voxelwise? A difference map is somewhat less sophisticated and give little to no information about significance.

Following to the reviewer suggestion, we have computed the statistical map (Figure 2c), where we represent the t-values in a colour scale ranging from 2.2 to 3.3 (equivalent to uncorrected p values ranging from 0.05 to 0.005). Also in this case, the map pointed to the corpus callosum as the most affected white matter tract. Indeed, when we performed the ROI analysis of the FA in the corpus callosum, we observed a significant reduction in the *Cyfp1*^{+/-} mice.

a. Were there any tests used to account for multiple comparisons in the DTI examination? Or was the difference map used for directing the findings, so it was deemed unnecessary?

The reviewer is correct, based on the rsfMRI data and the DTI differential map we directed our analysis to the corpus callosum for being the most affected and interesting region. Therefore, as a single t-test was performed for the CC, no multiple comparison was deemed necessary

b. An FA difference t-test p-value of 0.016 may not hold up to multiple comparisons. I'm assuming more was done as it was mentioned in the results that FA was globally decreased in white matter structures outside of the fimbria where there was an increase. If these comments are mentioned off of the difference map then they are not significant and then should not be stated as fact.

We apologize for not having clearly described the changes in FA. The reviewer is correct. Based on our observations of the differential map we reported on a globally decreased FA. However, the differences found in the voxelwise analysis don't hold up after multiple comparisons. We have now removed that sentence from the results section.

3) No diffusion differences were detected (MD, AD, RD), but yet there was an FA difference. In looking at figure 2c it is likely a small (insignificant increase in RD) that is leading to the drop in FA, consistent with the myelination differences found. This should be discussed.

There is indeed a slight increase in RD; we have expanded the results section to include this point.

4) More information about the sequence parameters are needed for both MRI sequences, especially the DTI sequence. The b-value of 800 s/mm² is also extremely low. Why was this chosen? Gradient constraints?

We thank the reviewer for this advice and we understand his/her point. The used a b-value of 800 s/mm² was based on previous studies in rodents by one of the co-authors of this manuscript and also recently described in (Anckaerts et al *Neurobiology of Disease*, 2019).

5) With all the delineated regions of interest and the registration procedure already done using deformation based morphometry. Why are there no volume measurements done? Was the corpus callosum smaller as well?

Following the reviewer's suggestion we have analysed the volume of the corpus callosum and included the results in the revised manuscript (Supplementary Figure 1c).

Reviewer #3 (Remarks to the Author):

Manuscript Number: Nature Communications manuscript 18-35561

Title: The autism and schizophrenia-associated protein CYFIP1 regulates bilateral brain connectivity

Bagni and colleagues have assessed brain connectivity by numerous imaging and electrophysiological metrics using a model of dysregulated *Cyfp1*. The authors present a strong case for *Cyfp1*'s role in numerous structural abnormalities. Secondary, they present a small component that suggests these *Cyfp* alterations in structures and physiology are key in behavioral and functional outcomes by 2 behavioral assays important for motoric functional phenotyping.

We thank the reviewer for the constructive feedback that helped us to improve the manuscript exploring more deeply how reduction of *Cyfp1* affects different ASD and SCZ-like behaviours.

- This sentence is awkward and incorrect as stated "...as the gene most likely causally associated with SCZ and ASD..." there is not a single gene that causes either disease and certainly associations are not causal.

- *Cyfp1* only receives a score of 3 by SFARI genome browser, meaning some suggestive evidence, but not very high so the importance to ASD was overstated.

We apologize for the unclear statement. We have now modified this part of the text as follows: "*single nucleotide polymorphisms (SNPs) and point mutations single out CYFIP1 as the gene of that region most likely associated with SCZ and ASD*".

- The illustration in 6A is unessential.

Following the reviewer's comment we have removed that illustration. Because we added several different behavioural tests we thought to make simpler schemes/images to guide the reader. We hope the reviewer is pleased with that.

- The two assays shown and tested are the bare minimum of behavioral testing for MOTOR dysfunction and lack of motor coordination. They tell you no other information. Since only simple motor outcomes can be concluded from these two assays, it was unimpressive for a "functional" connection.

We thank the reviewer for this comment and suggestions. We used the rotarod as it has been shown in mice with corpus callosum defects to result in reduced motor coordination (Yoo et al., *J Comp Neurol*, 2017. PMID: 28339102 ; Franco-Pons et al., *Toxicol Lett*, 2007. PMID: 17317045). The *Cyfp1*^{+/-} mice did not show striking deficits. We reasoned that the *Cyfp1*^{+/-} mice could implement over time a clumsy (but

still efficient) way of walking in order to stay on the rod. Following the reviewer's suggestions we have performed other motor-related assays that measure more fine-tuned motor coordination (ladder rung walking test) and muscle strength (hanging wire test). With the ladder rung walking test, we have additional evidence for a reduced motor coordination in the *Cytip1^{+/-}* mice (Figure 6f and g). In this task, mice have to walk over regularly spaced rungs, and the proportion of slips (i.e. wrongly placed paws) is measured. Notably, the *Cytip1^{+/-}* mice show more slips compared to WT mice (20% vs 6%). In addition, this impairment is consistent over all the trials. We did not observe any abnormalities in muscle strength using the hanging wire test (see Supplementary Figure 3d), from which we can exclude that muscle strength deficits account for the reduced motor coordination.

- It is unclear why, given that this model is a theoretical model of SCZ and ASD, no behavioral testing was performed that has translational relevance such as working memory for SCZ and/or social or lack of cognitive flexibility for ASD

[Redacted]

- In brief:

Concerning SCZ-relevant behaviours, we found no differences in working memory in two variants of the

Y-maze where *Cytip1* heterozygous mice show similar preference (forced alternation) and arm alternations (spontaneous alternation) as the WT controls (Supplementary Figure 2c and d). We also analysed the startle response with the pre-pulse inhibition (PPI) task. We found that *Cytip1^{+/-}* mice have reduced PPI compared to WT (Figure 7b). Similar effects were observed in patients with schizophrenia

[Redacted]

Another ASD-relevant behaviour is sensory processing. To analyse this, we tested the mice in the texture novel object recognition task (tNORT), where mice have to distinguish a familiar object from a novel

object. As mice are novelty seeking, they will spend more time with the novel object. *Cyfp1^{+/-}* mice show no preference for the novel texture, indicating reduced sensory discrimination (Figure 7a).

We have performed two independent behavioural assays to investigate cognitive flexibility (the Morris Water Maze [MWM], and the Touchscreen chambers). The *Cyfp1^{+/-}* mice are less precise in choosing the target quadrant during the reverse phase of the MWM. Because the observed differences are very minor, we conclude that the *Cyfp1^{+/-}* mice have no obvious defects in cognitive flexibility - measured with these two independent tests (Supplementary Figure 5).

Cyfp1^{+/-} mice were also tested for repetitive behaviours in the grooming and marble burying test. *Cyfp1^{+/-}* mice have similar grooming and burying behaviour as WT controls (Supplementary Figure 4a and b). In addition, while this work was under revision and posted in bioRxiv, Bachmann and colleagues (2019) observed as well motor defects in the *Cyfp1^{+/-}* mice and defects in sociability.

• The methods of running the rotarod are not standard **since 10 minute resting intervals** does not seem like substantial rest period.

We thank the reviewer for this comment and interestingly a large diversity of inter trial intervals (ITI) has been reported in the literature, ranging from just a few minutes up to 45 min. 10-15 minutes has been recommended to allow enough recovery time for the mouse after 5 min of intense activity. We have followed the protocol previously published by our collaborators in this manuscript with expertise in behaviour.

- Goddyn et al., Behav Brain Res, 2006. PMID: 16860407
- Denayer et al., Neurobiol Dis, 2008. PMID: 19118178
- Pooters et al., Behav Brain Res, 2016. PMID: 26548360
- Stroobants et al., Behav Brain Res, 2013. PMID: 23219967

Another published paper makes use of a ITI of 10min:

- Basu et al., 2009, Mol Psychiatry. PMID: 19065142

Moreover, several papers report even shorter ITI times:

- 1 min ITI: Kheirbek et al., J Neurosci, 2009. PMID: 19793969
- 2-5 min ITI: Franco-Pons et al., Toxicol Lett, 2007. PMID: 17317045
- 5 min ITI: Callahan & Abercrombie, Neurobiol Dis, 2015. PMID: 25772440

• There only seems to be significant difference at trial 1 and timepoint 1 which MEANS there is no difference in motor coordination. What is averaged in Panel 6B to detect this difference? This is not standard reporting and only further convinced me that there were not genotype differences in the behaviors tested.

We apologize for the lack of clarity. We have now rephrased the text to better explain what the graph represents. Figure 6b represents the average fall latency of the 4 trials per mouse. We agree with the reviewer that the main difference seems to be on trial 1. A two-way Repeated Measures ANOVA gave a statistically significant genotype effect, meaning that in general the genotypes are different. We have therefore performed an additional independent test to analyse motor coordination (see before: the ladder rung walking test). In this paradigm, *Cyfp1^{+/-}* mice show a more striking defect in motor coordination (Figure 6 c and d) further supporting our initial conclusions.

• For open field, I would not use distance moved but distance traveled, and horizontal and/or vertical activity counts, however there are clearly NO genotype differences. BUT 10 minutes in the open field is not standard and extremely short. Most behavioral cores use 30 minutes to 1 hour sessions to make determinations on motor differences.

Following the reviewer's suggestion we have now called it distance travelled and we have measured horizontal activity (walking/running). In the grooming test, we did not find any difference in vertical activity

(rearing). In most of the published literature the open field test is performed over 10 min (see references below). In addition, longer times could also account for motivation and exploration. To assess for motor defects we relied on the other motor behavioural tasks (see above).

Representative publications with 10 min open field exposure:

- Goddyn et al., Behav Brain Res, 2006. PMID: 16860407
- Perhoc et al., Neuropharmacology, 2018. PMID: 30571958
- Mehla et al., Neurobiol Aging, 2018. PMID: 30508733
- Lin et al., J Neurosci, 2018. PMID: 29279308

- To try and make conclusions about a functional role in disorders of the brain and behavior by rotarod and 1 open field metric is too far of stretch.

We agree with the reviewer and following his/her suggestion we have performed additional tests to better characterize the behavioural phenotype of the *Cyfp1^{+/-}* mice (see above).

- It is more likely that this effect was not present to begin with rather than that this small defect was ameliorated after repeated trials in rotarod. To exclude that the abnormal coordination was due to reduced locomotion.

To address this concern we have analysed motor coordination using the ladder rung walking test, which is more suitable to test motor coordination. We found a striking defect in the *Cyfp1^{+/-}* mice (Figure 6c and d).

- No core domains of the ASD phenotype were even tested.

[Redacted]

- There is no analysis of repetitive behaviors and no analysis of instance on sameness or preservative learning. Nor have they shown reciprocity in social behaviors like in a dyad social interaction.

[Redacted]

REVIEWERS' COMMENTS:

Reviewer #1 (Remarks to the Author):

The authors have added substantial additional data to the manuscript, in particular to show that the number of oligodendrocytes is not altered in CYFIP1 Het mice and to further characterize the behavioral phenotype of these mice. The mechanism by which CYFIP1 deficiency leads to the observed changes remains largely unknown, and the authors refer to future studies in which this will be addressed. Accordingly, this study encompasses an interesting but somewhat loosely connected set of phenotypes resulting from heterozygous deletion of CYFIP1.

One of the data sets added by the authors includes a texture novel object recognition paradigm, which the authors claim measures sensory perception. Can the authors rule out that the CYFIP1 Het mice have a general impairment in novelty-seeking behavior, rather than a specific sensory deficit? In the publication cited by the authors (Orefice et al. 2016), this is ruled out by comparison to a regular novel object recognition task (Figure 1D). If the authors did not perform this control, they should state in the manuscript that CYFIP1 Het mice have an impairment in sensory processing or novelty-seeking behavior, since they cannot distinguish between the two.

Reviewer #2 (Remarks to the Author):

The authors have address all of my comments and concerns. I appreciate the additional information provided including the new behavioural additions.

Jacob Ellegood

Reviewer #3 (Remarks to the Author):

The additions and explanations greatly improve the manuscript and conclusions. I have no major hesitations with endorsing its publication.

One small comment to the authors is that when performing behavioral neuroscience methodologies rather than relying on published literature from any laboratory with varying expertise, it makes a stronger case for protocol justification if behavioral protocols are followed as those published by behavioral neuroscience laboratories and in journals and texts such as Current Protocols.

There is limited application of ARRIVE guidelines and inconsistencies in baseline responses to controls, generally, therefore there is a rigor and reproducibility "crisis" in neuroscience. As a behaviorist, hoping to improve rigor, I wanted to outline this suggestion to the authors for the future study o this and other genetic mouse models.

Point-by-point Reviewers.

Dominguez-Iturza et al.

Reviewer #1 (Remarks to the Author):

The authors have added substantial additional data to the manuscript, in particular to show that the number of oligodendrocytes is not altered in CYFIP1 Het mice and to further characterize the behavioral phenotype of these mice. The mechanism by which CYFIP1 deficiency leads to the observed changes remains largely unknown, and the authors refer to future studies in which this will be addressed. Accordingly, this study encompasses an interesting but somewhat loosely connected set of phenotypes resulting from heterozygous deletion of CYFIP1.

One of the data sets added by the authors includes a texture novel object recognition paradigm, which the authors claim measures sensory perception. Can the authors rule out that the CYFIP1 Het mice have a general impairment in novelty-seeking behavior, rather than a specific sensory deficit? In the publication cited by the authors (Orefice et al. 2016), this is ruled out by comparison to a regular novel object recognition task (Figure 1D). If the authors did not perform this control, they should state in the manuscript that CYFIP1 Het mice have an impairment in sensory processing or novelty-seeking behavior, since they cannot distinguish between the two.

We thank the reviewer for the comments.

We have not performed the regular novel object recognition task. We therefore agree with the reviewer that the observed phenotype can be due to both sensory perception and/or novelty seeking (also affected in ASD patients). We have modified the results and discussion sections to clarify this point.

Reviewer #2 (Remarks to the Author):

The authors have address all of my comments and concerns. I appreciate the additional information provided including the new behavioural additions.

Jacob Ellegood

We thank Dr. Jacob Ellegood for his positive comments and suggestions that helped us to improve our manuscript.

Reviewer #3 (Remarks to the Author):

The additions and explanations greatly improve the manuscript and conclusions. I have no major hesitations with endorsing its publication.

One small comment to the authors is that when performing behavioral neuroscience methodologies rather than relying on published literature from any laboratory with varying expertise, it makes a stronger case for protocol justification if behavioral protocols are followed as those published by behavioral neuroscience laboratories and in journals and texts such as Current Protocols.

There is limited application of ARRIVE guidelines and inconsistencies in baseline responses to controls, generally, therefore there is a rigor and reproducibility "crisis" in neuroscience. As a behaviorist, hoping to improve rigor, I wanted to outline this suggestion to the authors for the future study of this and other genetic mouse models.

We thank the reviewer for the comments and advice.

Most of our behavioral protocols are based on published work by behavioral neuroscientists including our collaborators in this manuscript -Dr. Rudi D'Hooge and Dr. Zsuzsanna Callaerts-Vegh. We have also discussed our behavioral strategies with Dr. Leonardo Restivo – head of the behavioral platform in our Department.